# Effects of daily meteorology on the interpretation of space-based remote sensing of $NO_2$

Joshua L. Laughner[1], Azimeh Zare[1], and Ronald C. Cohen[1,2]

[1]Department of Chemistry, University of California, Berkeley, Berkeley, CA, USA
[2]Department of Earth and Planetary Science, University of California, Berkeley, Berkeley, CA, USA

*Correspondence to:* R.C. Cohen (rccohen@berkeley.edu)

**Abstract.** Retrievals of tropospheric $NO_2$ columns from UV/visible observations of reflected sunlight require a priori vertical profiles to account for the variation in sensitivity of the observations to $NO_2$ at different altitudes. These profiles vary in space and time but are usually approximated using models that do not resolve the full details of this variation. Currently, no operational retrieval simulates these a priori profiles at both high spatial and high temporal resolution. Here we examine the additional benefits of daily variations in a priori profiles for retrievals already simulating a priori $NO_2$ profiles at sufficiently high spatial resolution to identify variations of $NO_2$ within urban plumes. We show the effects of introducing daily variation into a priori profiles can be as large as 40% and $3 \times 10^{15}$ molec. cm$^{-2}$ for an individual day and lead to corrections as large as $-13\%$ for a monthly average in a case study of Atlanta, GA, USA. Additionally, we show that $NO_x$ emissions estimated from space-based remote sensing using daily, high spatial resolution a priori profiles are $\sim 100\%$ greater compared to a retrieval using spatially coarse a priori profiles, and 20–45% less compared to a retrieval using monthly averaged high spatial resolution profiles.

## 1 Introduction

$NO_x$ (= $NO$ + $NO_2$) is an atmospheric trace gas family that plays an important role in regulating the production of $O_3$ and particulate matter. $NO_x$ is emitted into the atmosphere by natural processes (e.g. lightning, biomass burning) and anthropogenic sources, notably combustion. Understanding the contribution of each source is vital to determining the effectiveness of current and future efforts to improve air quality and to understanding the chemistry of the atmosphere. Studies have utilized satellite observations to constrain $NO_x$ emissions from lightning (e.g. Miyazaki et al. 2014; Beirle et al. 2010; Martin et al. 2007; Schumann and Huntrieser 2007), biomass burning (e.g. Castellanos et al. 2014; Mebust and Cohen 2014, 2013; Miyazaki et al. 2012; Mebust et al. 2011), anthropogenic $NO_x$ emissions and trends (e.g. Ding et al. 2015; Lamsal et al. 2015; Tong et al. 2015; Huang et al. 2014; Vinken et al. 2014b; Gu et al. 2013; Miyazaki et al. 2012; Russell et al. 2012; Lin et al. 2010; Kim et al. 2009), soil $NO_x$ emissions, (e.g. Zörner et al. 2016; Vinken et al. 2014a; Hudman et al. 2012), and $NO_x$ lifetime (Liu et al., 2016; Lu et al., 2015; de Foy et al., 2014; Valin et al., 2013; Beirle et al., 2011).

The process of retrieving a tropospheric $NO_2$ column with UV/visible spectroscopy from satellites requires three main steps. First, the raw radiances are fit using Differential Optical Absorption Spectroscopy (DOAS) to yield slant column densities (Richter and Wagner, 2011). Then, the stratospheric $NO_2$ signal must be removed (Boersma et al., 2011; Bucsela et al., 2013). Finally, the tropospheric slant column density (SCD) must be converted to a vertical column density (VCD) by use of an air mass factor (AMF) and Eq. (1). Depending on the specific algorithm, $NO_2$ obscured by clouds may be ignored (producing a visible-only tropospheric $NO_2$ column, e.g. Boersma et al. 2002), corrected by use of an assumed ghost column (e.g. Burrows et al. 1999; Koelemeijer and Stammes 1999), or corrected via the AMF (e.g. Martin et al. 2002). In all cases, the AMF must account for the varying sensitivity of the satellite to $NO_2$ at different altitudes, and therefore a priori knowledge of that sensitivity and the vertical profile of $NO_2$ is required. Over low-reflectivity surfaces, light scattered in the atmosphere is the primary source of radiance at the detector. The probability of back-scattered light penetrating to a given altitude is greater for higher altitudes; thus there is greater interaction with, and therefore greater sensitivity to, $NO_2$ at higher altitudes (Richter

and Wagner, 2011; Hudson et al., 1995). Because of this, the correct AMF is smaller in locations influenced by surface $NO_x$ sources. The relative contribution of errors in the calculated sensitivity and in the a priori profiles of $NO_2$ to error in the final VCD varies between polluted and clean pixels (Boersma et al., 2004). Previous work (e.g. Russell et al. 2011) has sought to reduce errors in both, and highlighted the importance of accurate a priori profiles in urban areas.

$$\text{VCD} = \frac{\text{SCD}}{\text{AMF}} \tag{1}$$

A priori $NO_2$ profiles are generated using chemical transport models. Previous studies (e.g. Cohan et al. 2006, Wild and Prather 2006, Valin et al. 2011, Vinken et al. 2014b, Schaap et al. 2015) have demonstrated these modeled $NO_2$ profiles are strongly dependent on the spatial resolution of the chemical transport model used. The impact of model spatial resolution on satellite retrievals has been evaluated through case studies (Valin et al., 2011; Heckel et al., 2011; Yamaji et al., 2014) and through what could be termed "regional" retrievals (Russell et al., 2011; McLinden et al., 2014; Kuhlmann et al., 2015; Lin et al., 2015) that trade complete global coverage for improved spatial resolution of the input assumptions. These studies recommend model resolution of $< 20$ km to accurately capture $NO_x$ chemistry on a priori profiles. Russell et al. (2011) showed that increasing the spatial resolution of the input $NO_2$ profiles produces a retrieval that more accurately represents contrast in the spatial features of $NO_2$ plumes, reducing systematic bias by as much as 30%. Reducing these biases improves the clarity of the observed urban-rural gradients by providing unique urban and rural profiles, rather than one that averages over both types of locations. McLinden et al. (2014) showed that using 15 km resolution profiles increased the $NO_2$ signal of the Canadian oil sands by $\sim 100\%$ compared to the DOMINO and NASA SP products, which they state corrects a low bias in the retrieved column amounts.

Currently, only the Hong Kong-OMI retrieval has made use of daily a priori $NO_2$ profiles at $< 20$ km spatial resolution (Kuhlmann et al., 2015). Their retrieval covered the Pearl River Delta for the period October 2006 to January 2007. No operational retrieval covering the majority of the OMI data record does so. The current generation Berkeley High Resolution (BEHR) (Russell et al., 2011, 2012) and OMI-EC (McLinden et al., 2014) retrievals simulate monthly average $NO_2$ profiles at 12 and 15 km, respectively. Conversely, the DOMINOv2 (Boersma et al., 2011), POMINO (Lin et al., 2015), and DOMINO2_GC (Vinken et al., 2014b) retrievals simulate daily profiles at $3°$ lon $\times$ $2°$ lat (DOMINO) and $0.667°$ lon $\times$ $0.5°$ lat (POMINO and DOMINO2_GC), respectively, which is insufficient to capture the full spatial variability of $NO_2$ plumes, but does capture large scale variations in meteorology. Lamsal et al. (2014) quantitatively compared $NO_2$ average profile shapes measured from the P3-B aircraft for each of six sites in the DISCOVER-AQ Baltimore/DC campaign with the modeled profile shape from the GMI chemical transport model used to compute the $NO_2$ a priori profiles in the NASA Standard Product v2 retrieval, which uses monthly average $NO_2$ profiles at $2° \times 2.5°$ spatial resolution. They found up to 30% differences between the measured and modeled profile shape factors (i.e. $S(p)$ in Eq. 3) at any single pressure throughout the troposphere. Several sites (Edgewood, Essex, and Beltsville) had less $NO_2$ than the model throughout the free troposphere, and Edgewood also exhibited an elevated $NO_2$ layer at 970 hPa not captured in the model.

Lamsal et al. (2014) also noted that there was significant day-to-day variability in the measured profiles that cannot be captured by a monthly average model; however, they do not quantify those differences. These day-to-day differences can be significant in a priori $NO_2$ profiles. Valin et al. (2013) showed that the concentration of $NO_2$ downwind of a city increases significantly with wind speed, observing that $NO_2$ 100–200 km downwind from Riyadh, Saudi Arabia was approximately 130–250% greater for wind speeds between 6.4–8.3 m s$^{-1}$ than wind speeds $< 1.9$ m s$^{-1}$. When monthly average a priori profiles are used, this is not accounted for in the retrieval. The effect on the AMF is illustrated in Fig. 1c. Compared to the monthly average a priori profiles, daily profiles from a day with fast winds would contain greater near-surface $NO_2$ further from the city. As discussed before Eq. (1), UV/visible satellite observations of $NO_2$ are less sensitive to $NO_2$ at low altitudes, so this requires smaller AMFs at a greater distance from the city on days with fast winds to compensate through Eq. (1).

These day-to-day variations may be particularly important for methods such as Beirle et al. (2011), Valin et al. (2013), Lu et al. (2015), and Liu et al. (2016) that use observations sorted by wind speed to derive detailed information about $NO_x$ chemistry and emissions from space-borne observations. This is a very valuable tool because of the wealth of data available from OMI (Levelt et al., 2006) and expected from upcoming instruments such as TROPOMI (Veefkind et al., 2012), TEMPO (Chance et al., 2013), Sentinel-4 (Ingmann et al., 2012), and GEMS (Bak et al., 2013; Choi and Ho, 2015). However, the act of sorting data by wind speed transforms errors in the profile shape resulting from day-to-day variability in wind speed from random to systematic. For example, Beirle et al. (2011), Valin et al. (2013), and Lu et al. (2015) derive an effective $NO_x$ lifetime using data with fast wind speed, and Liu et al. (2016) does so by fitting a function with a component derived at slow wind speeds to data derived from days with fast wind speeds. On a day when the wind speed is faster than average, a priori $NO_2$ profiles taken from a monthly average model would have less near-surface $NO_2$ further from the city than is actually present for that day (i.e. Fig. 1c vs. 1a). The resulting incorrect AMFs would lead to an underestimation of the spatial extent of the plume, and could lead to an underestimate of the $NO_x$ lifetime as a consequence.

In this paper we explore how day-to-day changes in the a priori $NO_2$ profiles affect satellite retrievals of urban $NO_2$. Several scenarios are illustrated in Fig. 1. In each case the change in the AMF results because, over low albedo surfaces, a UV/visible satellite spectrometer is less sensitive to near surface trace gases, necessitating a smaller AMF to account for the reduced sensitivity. In Fig. 1a, the monthly average $NO_2$ plume is shown as the grayscale gradient, to emphasize that it is static from day to day. Most of the plume follows the prevailing wind direction (here, to the right), but because days with different wind directions are averaged together, there is some influence of the plume upwind of the city. Figure 1b shows a case where the daily winds are similar to the monthly average. This leads to a similar $NO_2$ plume as in the monthly average, but because we are not averaging different wind directions, the upwind plume influence is removed (increasing the AMF, reflecting the reduction in near-surface $NO_2$) and conversely the downwind AMFs are slightly smaller, due to a slight increase in near-surface $NO_2$ from not averaging in days when the wind direction is different. Figure 1c shows a case where the daily winds are faster than the average. Here the AMFs within the city need to be larger, as near-surface $NO_2$ is being removed more efficiently and transported downwind, where the AMFs must therefore be smaller. Finally, Fig. 1d has the wind change direction from the monthly average. Left of the city must have smaller AMFs to account for the presence of the plume not seen in the monthly average, and the opposite change occurs to the right.

We combine the high spatial resolution a priori previously developed as part of the BErkeley High Resolution (BEHR) algorithm (Russell et al., 2011) with high temporal resolution to demonstrate the impact of day-to-day variations in the modeled $NO_2$ profiles on the calculated AMFs surrounding a major urban area such as Atlanta, GA, USA. Atlanta provides an example of a strong $NO_x$ area source relatively isolated from other sources, with straightforward response of the day-to-day a priori profiles to meteorological variables. Our point is not to derive exact answers for the size and frequency of the effects of daily profiles, but rather to illustrate that these effects are large enough that their role should be assessed in any future analysis that does attempt to interpret space-based remote sensing of $NO_x$. We show that the variability in the a priori profiles is largely due to changes in wind speed and direction. We first consider the effects of day-to-day variations in a priori profile on AMFs for the region surrounding Atlanta for a fixed grid of OMI pixels, simplifying day-to-day comparisons. We then fully implement 91 days of retrieval to examine the effect on both day-to-day and monthly average $NO_2$ columns. Finally, we apply the exponentially-modified Gaussian (EMG) fitting method of Lu et al. (2015) to the new retrieval and show that the spatial and temporal resolution of the a priori profiles can significantly alter the derived emission rate and lifetime.

## 2  Methods

### 2.1  The Ozone Monitoring Instrument

The Ozone Monitoring Instrument (OMI), onboard the Aura satellite, is a polar-orbiting, nadir-viewing, UV/visible spectrometer with a swath width of 2600 km and a pixel size at nadir of $13 \times 24$ km$^2$. It observes backscattered solar radiation in the range of 270–500 nm with an average spectral resolution of 0.5 nm. (Levelt et al., 2006). It has a continuous data record since 1 Oct 2004, with global daily coverage for the first $\sim 3$ years of operation. Since 25 June 2007, anomalous radiances have been observed in several of the pixel rows. These have been classified as the "row anomaly" (http://projects.knmi.nl/omi/research/product/rowanomaly-background.php). As of 5 July 2011, one-third of OMI pixels are flagged as affected by the row anomaly, indicating that data from these pixels should not be used. Using only the pixels unaffected by the row anomaly, it takes two days to observe the entire globe. There are two publicly available global $NO_2$ products, the KNMI DOMINO product (Boersma et al., 2011) and the NASA Standard Product (Bucsela et al., 2013).

### 2.2  BErkeley High Resolution (BEHR) Retrieval

The BEHR retrieval is described in detail in Russell et al. (2011), and updates are described on the BEHR website (http://behr.cchem.berkeley.edu/Portals/2/Changelog.txt). The product is openly available for download at http://behr.cchem.berkeley.edu/. Briefly, the BEHR retrieval is based on the NASA Standard Product v2 (SP v2) retrieval (Bucsela et al., 2013). The total slant column densities (SCDs) are from OMNO2A v1.2.3 (Boersma et al., 2002; Bucsela et al., 2006, 2013), and have been recently evaluated by van Geffen et al. (2015) and Marchenko et al. (2015). The stratospheric subtraction and destriping used is that of the NASA SP v2 retrieval. The tropospheric AMF is then recalculated similarly to the AMF formalism described in Palmer et al. (2001). Clear and cloudy AMFs are calculated as shown in Eq. (2). $p$ represents the vertical coordinate as pressure. $w(p)$

represents scattering weights derived from the NASA SP v2 look up table. $g(p)$ represents the mixing ratio $NO_2$ a priori profile taken from WRF-Chem, simulated at 12 km resolution in the published BEHR product. $p_0$ represents the surface pressure (clear sky AMF) or cloud pressure (cloudy AMF) of the satellite pixel, and $p_{tp}$ the tropopause pressure. The cloud pressure is that provided in the NASA SP v2 product, and is retrieved using the OMI $O_2$-$O_2$ cloud algorithm (Acarreta et al., 2004; Sneep et al., 2008; Bucsela et al., 2013). A static tropopause pressure of 200 hPa is used. $p_{surf}$ in Eq. (3) is the terrain surface pressure. The integration is carried out using the scheme described in Ziemke et al. (2001) which allows integration of mixing ratio over pressure.

$$\text{AMF} = \int_{p_0}^{p_{tp}} w(p)S(p)\,dp \tag{2}$$

where

$$S(p) = \frac{1}{\int_{p_{surf}}^{p_{tp}} g(p)\,dp} g(p) \tag{3}$$

The scattering weights, $w(p)$, depend on the viewing geometry, surface albedo, and terrain pressure altitude. The BEHR algorithm uses the $0.05° \times 0.05°$ combined MODIS MCD43C3 black-sky albedo product and a surface pressure derived from the Global Land One-km Base Elevation project database (http://www.ngdc.noaa.gov/mgg/topo/globe.html; Hastings and Dunbar 1999) with a 7.4 km scale height as inputs to the clear sky scattering weights. Cloudy scattering weights treat the cloud pressure as the surface pressure and use an assumed cloud albedo of 0.8 (Stammes et al., 2008; Bucsela et al., 2013). The final AMF is computed as the cloud radiance fraction ($f_{rad}$) weighted average of the clear and cloudy AMFs (Eq. 4). The cloud radiance fraction is taken from the SP v2 data product (Bucsela et al., 2013).

$$\text{AMF}_{total} = f_{rad}\text{AMF}_{cloudy} + (1 - f_{rad})\text{AMF}_{clear} \tag{4}$$

Calculating clear and cloudy AMFs and using the weighted average to compute the final AMF is consistent with the OMI algorithm theoretical basis document (Boersma et al., 2002) and yields only the visible $NO_2$ column as the final product; the visible column is the value provided in the BEHRColumnAmountNO2Trop field. A scaling factor is provided in the BEHR product for users who wish to include the ghost column. This factor, $G$, is computed as:

$$G = \frac{V_{surf}}{(1 - f_{geo})V_{surf} + f_{geo}V_{cld}} = \frac{\int_{p_{surf}}^{p_{tp}} g(p)\,dp}{(1 - f_{geo})\int_{p_{surf}}^{p_{tp}} g(p)\,dp + f_{geo}\int_{p_{cld}}^{p_{tp}} g(p)\,dp} \tag{5}$$

where $V_{surf}$ and $V_{cld}$ are the modeled vertical column densities above the ground surface and cloud, respectively, and which are obtained by integrating the a priori profile above the surface or cloud pressure. $f_{geo}$ is the geometric cloud fraction included in the NASA standard product, which is the OMI $O_2$-$O_2$ cloud product (Acarreta et al., 2004). This factor is stored in the

BEHRGhostFraction field of the BEHR product. Multiplying the VCDs stored in BEHRColumnAmountNO2Trop by these values will provide the estimated total (visible + ghost) column.

The results obtained in this work use the visible columns only. The ghost column is not added in for any of the following results.

## 2.3 WRF-Chem

Modeled $NO_2$ a priori profiles are simulated using the WRF-Chem model v3.5.1 (Grell et al., 2005). The domain is 81 (east-west) by 73 (north-south) grid cells centered on 84.35° W, 34.15° N on a Lambert Conformal map projection (approximate

edges of the domain are 89.5° W–79.2° W and 30.3° N–38° N). Meteorological initial and boundary conditions are driven by the North American Regional Reanalysis (NARR) dataset. Anthropogenic emissions are taken from the National Emissions Inventory 2011 (NEI11) and scaled to 88.9% to account for 2011–2013 $NO_x$ reductions (EPA, 2016); total emissions of NO for the domain are approximately $3.1 \times 10^6$ kg NO day$^{-1}$. The MEGAN model (Guenther et al., 2006) is used to determine biogenic emissions. Chemical initial and boundary conditions for the domain are obtained from the MOZART chemical model

(Emmons et al., 2010). The RACM2 (Goliff et al., 2013) and MADE-SORGAM schemes are used to simulate gas-phase and aerosol chemistry respectively; the RACM2 scheme is customized to reflect recent advancements in understanding of alkyl nitrate chemistry using Browne et al. (2014) and Schwantes et al. (2015) as a basis. Lightning $NO_x$ emissions were inactive.

The model is run from 27 May to 30 August, 2013. Similar to Browne et al. (2014), the five day period 27–31 May is treated as a spin up period, thus we use 1 June to 30 August as our study time period. Model output is sampled every half hour; the

two output files from the same hour (e.g. UTC 1900 and 1930) are averaged to give a single hourly set of profiles. These hourly $NO_2$ profiles are used as the a priori $NO_2$ profiles in the BEHR retrieval (Section 2.2). To produce monthly average profiles, each hourly profile is weighted according to Eq. (7), where $l$ is the longitude of the profile and $h$ is the hour (in UTC) that WRF calculated the profile for. The weights are clamped to the range $[0, 1]$. These are used as the weights in a temporal average over the month in question. This weighting scheme gives higher weights to profiles closest to the OMI overpass time around 1330

local standard time.

$$w_l = 1 - |13.5 - (l/15) - h| \qquad (6)$$
$$w_l \in [0, 1]$$

The weighting scheme in Eq. (7) was chosen over simply using the model output for 1400 local standard time for each longitude to create smooth transitions between adjoining time zones. This attempts to account for the day-to-day variability in

OMI overpass tracks as well as the fact that pixels on the edge of a swath can be observed in two consecutive overpasses at different local times. More detail is given in the supplement.

A spatial resolution of 12 km is used as the high spatial resolution a priori. To determine the effect of coarser spatial resolution, the model is also run at 108 km resolution. At 12 km resolution, profiles are spatially matched to OMI pixels by averaging all profiles that fall within the pixel bounds. At 108 km resolution, the profile closest to the pixel is used. When using

daily profiles, they are temporally matched by identifying those closest to the scan time defined in the Time field of the NASA SP v2 data product.

## 2.4   Implementation of daily profiles

Two retrievals are used to study the effects of incorporating daily a priori profiles in the BEHR algorithm. The first is what we term a "pseudo-retrieval." To create this retrieval, an $11 \times 19$ (across $\times$ along track) subset of pixels from OMI orbit 47335 centered on the pixel at $84.2513°$ W and $33.7720°$ N is used to provide the pixel corners, solar and viewing zenith and azimuth

angles, terrain pressure, and terrain reflectivity. This swath places Atlanta near the nadir view of the OMI instrument (therefore providing pixels with good spatial resolution) while also remaining outside the row anomaly. This same subset of pixels is used for all days in the pseudo-retrieval. Cloud fractions are set to 0 for all pixels to consider clear-sky AMFs and simplify the pseudo-retrieval. AMFs are calculated for this subset of pixels with WRF-Chem $NO_2$ profiles from 1 June to 30 Aug 2013 in Eq. (2). This pseudo-retrieval will allow a simplified discussion of the effects of daily a priori profiles by:

1. Using a fixed set of OMI pixels. Because OMI pixels do not align day-to-day, using each day's true pixels makes a day-to-day comparison more difficult to see. In this pseudo-retrieval, that is alleviated.

2. Using a fixed set of OMI pixels also keeps the scattering weights ($w(p)$ in Eq. 2) constant as the parameters that the scattering weights depend on (solar and viewing zenith angles, relative azimuth angles, terrain albedo, and terrain height) are fixed.

3. Setting cloud fractions to 0 ensures that the AMF for every pixel is calculated with the full a priori profile, rather than just the above cloud part. Day-to-day variations in cloud fraction also lead to large changes in AMF because the presence of clouds changes both the scattering weights (due to high assumed reflectivity of clouds and smaller effective surface pressure compared to ground) while also obscuring the $NO_2$ profile below the cloud.

Essentially, the pseudo-retrieval is a idealized experiment in which we hold all other variables except the a priori profile

constant to compute the theoretical magnitude of the effect of using daily a priori profiles on the AMF. It will be used in Sect. 3.1 to demonstrate the effect of incorporating daily a priori profiles. The daily a priori profiles are also implemented in the full BEHR retrieval (no longer using a fixed set of pixels or forcing cloud fractions to 0) to determine the impact of including daily a priori profiles on the VCDs in a realistic case. When averaging in time, all pixels are oversampled to a $0.05° \times 0.05°$ grid. The contribution of each pixel is weighted by the inverse of its area.

## 2.5   Evaluation of exponentially-modified Gaussian (EMG) fits

Lu et al. (2015) and Valin et al. (2013) used $NO_2$ data from the DOMINO retrieval to study $NO_x$ emissions and lifetime from space, accounting for the effects of wind speed variation. To evaluate the impact of the a priori resolution on methods such as these, a similar procedure to fit an exponentially modified Gaussian function to $NO_2$ line densities is used. The surface wind direction and speed are calculated as the average of the first five layers ($\sim 500$ m) of the 9 WRF 12 km grid cells closest to

Atlanta at 1400 local standard time for each day. WRF wind fields are given relative to the model grid; however, the $x$ and $y$ coordinates of the grid do not correspond directly to longitude and latitude. Therefore, the wind fields must be transformed from grid-relative to earth-relative (http://www2.mmm.ucar.edu/wrf/users/FAQ_files/Miscellaneous.html) as:

$$U_{\mathrm{earth}} = U_{\mathrm{model}} \times \cos(\alpha) - V_{\mathrm{model}} \times \sin(\alpha) \tag{7}$$

$$V_{\mathrm{earth}} = V_{\mathrm{model}} \times \cos(\alpha) + U_{\mathrm{model}} \times \sin(\alpha) \tag{8}$$

where $U$ and $V$ are the longitudinal and latitudinal wind fields, and $\cos(\alpha)$ and $\sin(\alpha)$ are outputs from WRF as the variables COSALPHA and SINALPHA.

As in Valin et al. (2013), the satellite pixels are rotated so that wind direction (and therefore $NO_2$ plumes) for each day lie along the $x$-axis. Pixels affected by the row anomaly or with a cloud fraction $> 20\%$ are removed. Pixels within $1°$ upwind and $2°$ downwind are gridded to $0.05° \times 0.05°$ and integrated across $1°$ perpendicular to the $x$-axis. This produces line densities, which are a one-dimensional representation of the $NO_2$ concentration at various distances downwind of the city. Three a priori sets are used to create the retrievals used in this section: coarse (108 km) monthly average, fine (12 km) monthly average, and fine (12 km) daily profiles.

We use the form of the EMG function described in Lu et al. (2015) to fit the calculated $NO_2$ line densities, after expanding the definition of the cumulative distribution function:

$$F(x|a, x_0, \mu_x, \sigma_x, B) = \frac{a}{2x_0} \exp\left(\frac{\mu_x}{x_0} + \frac{\sigma_x^2}{2x_0^2} - \frac{x}{x_0}\right) \mathrm{erfc}\left(-\frac{1}{\sqrt{2}}\left[\frac{x - \mu_x}{\sigma_x} - \frac{\sigma_x}{x_0}\right]\right) + B \tag{9}$$

where erfc is the error function complement, i.e. $\mathrm{erfc}(x) = 1 - \mathrm{erf}(x)$. $F(x|a, x_0, \mu_x, \sigma_x, B)$ serves as an analytical function that can be fitted to the observed line densities. We find the values of $a$, $x_0$, $\mu_x$, $\sigma_x$, and $B$ that minimize the sum of squared residuals between $F(x|a, x_0, \mu_x, \sigma_x, B)$ and the line densities, $NO_2(x)$:

$$Resid(a, x_0, \mu_x, \sigma_x, B) = \sum_x \left(F(x|a, x_0, \mu_x, \sigma_x, B) - NO_2(x)\right)^2 \tag{10}$$

Eq. (10) is minimized using an interior-point algorithm, finding the values of $a$, $x_0$, $\mu_x$, $\sigma_x$, and $B$ that best fit the line densities. The values of $a$, $x_0$, $\mu_x$, $\sigma_x$, and $B$ have physical significance and so their optimum values yield information about the $NO_x$ emission and chemistry occurring within the plume (Beirle et al., 2011; de Foy et al., 2014; Lu et al., 2015). Specifically:

 – $a$ describes the total amount of $NO_2$ in the plume (referred to as the burden)

 – $x_0$ is the distance the plume travels in one lifetime, $\tau$. It relates to $\tau$ by $x_0 = \tau \times w$, where $w$ is wind speed.

 – $u_x$ describes the effective center of the emission source. In the supplement to Beirle et al. (2011), it is represented by $X$ which is the point at which exponential decay of the $NO_2$ plume begins.

– $\sigma_x$ is the standard deviation of the Gaussian component of the EMG function. Lu et al. (2015) terms this a "smoothing length scale," which describes smoothing of the data due to the spatial resolution and overlap of OMI pixels (Boersma et al., 2011). It can also be thought of as capturing effects of both the spatial extent of emissions and the turbulent wind field.

– $B$ is the background line density.

For each parameter, uncertainty from the fitting process itself is computed as the 95% confidence interval calculated using the standard deviation obtained from the fitting process. This is combined in quadrature with 10% uncertainty due to across wind integration distance, 10% uncertainty due to the choice of wind fields, and 25% uncertainty from the VCDs, similar to Beirle et al. (2011) and Lu et al. (2015). Technical details of the EMG fitting and uncertainty calculation are given in the supplement.

## 3 Results

### 3.1 Daily variations

Fig. 2 shows the average wind and modeled $NO_2$ columns for June 2013, and the AMF values for the psuedo-retrieval around Atlanta, GA, USA. Atlanta was chosen as the focus of this study because it represents a strong $NO_x$ source relatively isolated from other equally large sources. This ensures that changes to the a priori profiles on a daily basis can be attributed to a local cause. The prevailing wind pattern advects $NO_2$ to the northeast of Atlanta (the location of Atlanta is marked by the star), as can be seen in the wind field shown in Fig. 2b and the WRF-Chem $NO_2$ columns in Fig. 2c. The average surface wind speed over Atlanta for June is 5.0 m/s. This distribution of $NO_2$ leads directly to the lower AMFs seen to the northeast of Atlanta in Fig. 2d through Eq. (2).

To illustrate the effect of incorporating daily a priori profiles into the retrieval, we consider two days: 18 and 22 June 2013. These provide an illustration of the effect of changes in both wind speed and direction. Figure 3a–c shows the result from implementing the daily profiles for 22 June. On this day, the winds over Atlanta blow out of Atlanta to the northwest, with a speed at the surface of 4.5 m/s. This is similar to the monthly average speed (5.0 m s$^{-1}$) but are rotated 90° counterclockwise compared with the monthly average. The change in direction results in much greater near-surface $NO_2$ to the northwest compared to the monthly average (Fig. 3b) as the wind direction advects $NO_2$ into an area with low $NO_2$ in the monthly average.

Figure 3c shows that the greater near-surface $NO_2$ to the northwest results in lower AMFs than average (red), while the opposite is true to the east (blue). The greater near-surface $NO_2$ in profiles to the northwest weights $S(p)$ in Eq. (2) more heavily towards lower altitudes, where $w(p)$ is less, thus decreasing the overall AMF by ~15%. The increase in AMFs to the east reflects the inflow of cleaner air from the shift in winds. This reduces near-surface $NO_2$ and increases the weight of higher altitudes of $S(p)$, increasing the AMFs by ~10–35% (the colorbar saturates at ±25% to make the decrease to the northwest easier to see).

Wind speed also plays an important role in determining the a priori profile shape through transport and chemistry. Fig. 3d–f shows results from 18 June, where the wind speed over Atlanta averaged 9.1 m/s. This results in faster advection away from emission sources, with 10–15% increases in modeled $NO_2$ columns to the west as the plume is driven east more strongly. The greatest decreases in AMF (and thus increases in VCD) are as much as $-13\%$ and occur between $84°$ and $83°$ W where the increased wind speed has advected the $NO_2$ plume farther than the average. There is also a 2–13% increase along the east edge of Atlanta, resulting from the shift of the plume center east.

When the change in AMF from using the hybrid daily a priori profiles is averaged over the full time period studied (1 June–30 Aug), the percent change in AMF is on average +3.6% throughout the domain with a maximum of +9.8%. All pixels show a positive change. This occurs because 77% of the daily profiles have less $NO_2$ than the corresponding monthly average profile, as most pixels will be upwind from the city on any given day and will see a decrease in $NO_2$ when upwind and downwind days are no longer averaged together. This reduces the denominator in Eq. (3) and increases the contribution of upper tropospheric scattering weights to the AMF. Scattering weights increase with altitude; therefore, this results in a systematic increase of the AMF throughout the domain for the pseudo-retrieval.

We also consider the relative importance of day-to-day changes in the boundary layer of the a priori profiles versus day-to-day changes in the free troposphere of the a priori profiles by running the pseudo-retrieval with a set of hybrid daily profiles that only include day-to-day variability below 750 hPa and use a monthly average profile above that. The changes in AMFs using these hybrid profiles versus monthly average profiles are very similar to those observed when using the full daily profiles. In general, the hybrid profiles has a slightly greater average increase in AMFs (+3.2% vs. +2.7%) and slightly less extreme changes, but the overall distribution of changes in AMFs is very similar. From this, we can conclude that changes in the boundary layer of the a priori profiles are the dominant reason for changes to the AMFs. However, the WRF-Chem simulations used to produce the a priori profiles did not include lightning $NO_x$, so this should be considered a lower bound for the effect of day-to-day changes in the free troposphere. The detailed comparison is described in the supplement.

## 3.2  Effects on retrieved vertical column densities in full retrieval

To determine the effect the inclusion of daily a priori profiles has on the final retrieved vertical column densities (VCDs), the daily profiles were implemented in the full BEHR retrieval. Effects on individual days and multi-month average VCDs are presented here. The cities of Birmingham, AL, USA and Montgomery, AL, USA are included to demonstrate that this effect is significant for cities of various sizes. Atlanta, GA, USA is the largest with approximately 5.7 million people, followed by Birmingham, AL, USA with 1.1 million, and Montgomery, AL, USA with 374,000 (United States Census Bureau).

Table 1 describes how frequently significant changes in the retrieved VCD occur for pixels within 50 km of Atlanta, Birmingham, and Montgomery. Changes are considered significant by two different criteria. First, we consider the global mean clear-sky uncertainty from Bucsela et al. (2013). As we are modifying the a priori profiles, and thus potentially the uncertainty associated with the choice of profiles, this gives us a fixed value to compare against. Second, we use the quadrature sum of uncertainties from spectral fitting ($0.7 \times 10^{15}$ molec. cm$^{-2}$, Boersma et al. 2007, 2011), stratospheric separation ($0.2 \times 10^{15}$ molec. cm$^{-2}$, Bucsela et al. 2013), and AMF calculation (20%, Bucsela et al. 2013), assuming that these are independent

and so can be added in quadrature (Boersma et al., 2004). We consider the fraction of days with at least one pixel exhibiting a significant change in VCD (rather than the fraction of pixels) because the main $NO_2$ plume may only fall within a small

number of pixels. Up to 54% of days exhibit changes in the VCDs greater than $1 \times 10^{15}$ molec. cm$^{-2}$, and up to 43% exhibit changes greater than the quadrature sum of uncertainties. This indicates that when considering individual daily measurements, a considerable fraction of days with any valid pixels would have biases in the retrieved VCDs above the uncertainty due to the temporal resolution of the a priori $NO_2$ profiles.

For both significance criteria, Table 1 also indicates that Birmingham and its surrounding area exhibits the largest and most

frequent changes when using a daily a priori profile. Figure 4a shows the NO emissions throughout this domain. Birmingham has the second largest $NO_x$ emission rate, after Atlanta, while Montgomery has the smallest of the three cities considered. We note that the largest changes are not associated with the city with the greatest $NO_x$ emissions. Both Atlanta and Birmingham fall entirely within the $NO_x$ suppressed regime in the model, so the larger changes in Birmingham are not because $NO_x$ chemistry transitions between the $NO_x$ suppressed and $NO_x$ limited regimes. Instead, the magnitude of these changes is due to

Birmingham's intermediate size, where significant $NO_2$ is present, but emission occurs over a small enough area that changes in wind direction can significantly affect $NO_2$ concentration at a short distance from the source. When considering changes to be significant if they exceed $1 \times 10^{15}$ molec. cm$^{-2}$, Montgomery has the least frequent significant changes because it has the smallest VCDs, so a change to the AMF needs to be rather large to produce a significant change in the VCD by this metric, since the AMF is a multiplicative factor. When considering the quadrature sum of errors as the significance criterion,

Montgomery and Atlanta both demonstrate significant changes $\sim 20\%$ of the time.

Implementing the daily profiles also changes the average VCDs, in addition to the day-to-day changes in VCDs discussed above. Figure 4b shows the changes in VCDs averaged over the period studied. The largest decrease around Atlanta is to the northeast, along the direction that the monthly average model results placed the $NO_2$ plume, but clear decreases can also be seen to the northwest and southwest. In these directions, a systematic decrease of up to 8% ($4 \times 10^{14}$ molec. cm$^{-2}$) is observed.

Although this change is small, it is expected to be systematic. Statistically, a pixel's a priori profile is more likely to have less surface $NO_2$ when different wind directions are no longer averaged in, thus decreases in the VCD when using a daily a priori profile are more common.

Greater relative changes are observed around the smaller cities of Birmingham (down to $-12.9\%$, $5 \times 10^{-14}$ molec. cm$^{-2}$) and Montgomery (down to $-13\%$, $4 \times 10^{-14}$ molec. cm$^{-2}$). This appears to be due primarily because the areas of emissions

are smaller which makes shifts in wind direction have a greater average relative effect on the plume shape.

We also compare this average change to the measurement uncertainty. The uncertainty due to random errors in the retrieval should reduce as the square root of the number of observations, but delineating random and systematic errors in the retrieval is challenging (Boersma et al., 2004). The most optimistic approach assumes that the global average uncertainty of $1 \times 10^{15}$ molec. cm$^{-2}$ (Bucsela et al., 2013) can be treated entirely as random error, and can be reduced by $\sqrt{40}$ for the number of observations (not impacted by clouds or the row anomaly), to a lower bound of $\sim 1.6 \times 10^{14}$ molec. cm$^{-2}$. Most of the changes near the three cities exceed this lower limit. More realistically, the spectral fitting and stratospheric uncertainty may be considered largely random, but only part of the error in the AMF calculation is random, due to spatial or temporal auto-

correlation in the models or ancillary products (Boersma et al., 2004). For simplicity, we assume that the spectral fitting and stratospheric subtraction errors are entirely random, while only half of the error in the AMF is random. This reduces the error from $\sqrt{(0.7 \times 10^{15})^2 + (0.2 \times 10^{15})^2 + (20\%)^2}$ to $\sqrt{(0.11 \times 10^{15})^2 + (0.03 \times 10^{15})^2 + (11.6\%)^2}$. Only the largest changes near Birmingham and Montgomery exceed this threshold. This more conservative estimate suggests that the changes in averages are primarily important for smaller or very geographically concentrated cities, where wind direction can have a large effect. Nevertheless, larger cities may exhibit important changes as well.

Unlike the pseudo-retrieval, where we only allowed the a priori profiles to vary day-to-day and clouds were set to zero, there is a some spatial structure to these average changes. This is primarily a statistical phenomenon. We use only pixels with cloud fraction $< 20\%$, which reduces the number of pixels in the average. Within this subset, the wind blows to the southeast out of Atlanta more frequently than other directions; so the increases due to properly accounting for the presence of surface $NO_2$ average with the more typical decreases to give a small average change. The other directions exhibit the expected average decrease in VCDs due to the average increase in AMFs discussed in section 3.1. We expect that over longer periods of time all directions would see a 2–6% decrease in the average VCDs.

## 4 Discussion

### 4.1 Importance of model uncertainty

WRF-Chem has generally been found to reproduce wind fields, especially above 2 m s$^{-1}$ (Tie et al., 2007; Zhang et al., 2009), and spatial variability of trace gases (Follette-Cook et al., 2015) well. Nevertheless, a natural concern when modeling daily $NO_2$ profiles for satellite retrievals is the accuracy of the plume location. We, however, note that the transition from monthly average to daily profiles does not necessarily result in increased model uncertainty, but rather a change in the type of uncertainty.

When using monthly average profiles, the uncertainty in the modeled $NO_2$ concentrations compared to the true mean will be reduced (assuming at least some component of the error is random in nature), but the true day-to-day variability not captured by the monthly average effectively becomes a new error term. In contrast, when using daily profiles, the random model error is not reduced, but the day-to-day variability is also not averaged out. Ideally, the error in a set of daily profiles will manifest as deviation from the true set of profiles for that day, rather than the monthly profiles' smaller deviation from a mean set of profiles that itself may not represent any single day.

An important step in managing the uncertainty in the daily profiles is to constrain the modeled meteorology with observations or reanalysis datasets. By default, meteorology in WRF is constrained via initial and boundary conditions only. With larger domains and longer runs, further constraints using four-dimensional data assimilation (FDDA, Liu et al. 2006) and/or objective analysis (Follette-Cook et al., 2015; Wang et al., 2014; Yegorova et al., 2011), possibly combined with periodic model reinitialization (Otte, 2008) are strongly recommended.

## 4.2 Effects on space-based lifetime and emissions constraints

Recently several authors have used wind-sorted satellite $NO_2$ observations to probe $NO_x$ chemistry and emissions from space (Beirle et al., 2011; Valin et al., 2013; de Foy et al., 2014; Lu et al., 2015; Liu et al., 2016). We apply the EMG fitting method of Lu et al. (2015) to $NO_2$ line densities derived from $NO_2$ columns retrieved using the daily and monthly average a priori

profiles, as well as a monthly average profile simulated at 108 km resolution for both Atlanta and Birmingham. To match the method of Lu et al. (2015) as closely as possible, we use 3 m s$^{-1}$ as the division between slow and fast winds.

We acknowledge that a 91 day averaging period is significantly shorter than those used in Beirle et al. (2011), Valin et al. (2013), or Lu et al. (2015) (5 years, summer half-year for 7 years, and summer half-year for 3 year periods, respectively). However, since the goal of this section is to compare the results obtained using three different sets of a priori profiles with all

other variables equal, we believe that 91 days is sufficient for this purpose.

Additionally, we do not include days around Atlanta in which the wind blows towards the southeast (specifically 0° to -112.5°, 0° is defined as east, negative values are clockwise from east). Significant suburban $NO_2$ columns near 83.5° W, 33° N add a secondary maximum to the line densities which can erroneously lengthen the decay time of the fit. All wind directions are used for Birmingham.

Accounting for the spatial and temporal variability of $NO_2$ in the a priori profiles leads to several notable changes in the line densities and the resulting EMG fits. Figure 5 shows the line densities and the corresponding EMG fits around Atlanta for the average over the 91 day study period. Table 2 enumerates the values obtained for the fitting parameters in Eq. (9) for the fits of the Atlanta $NO_2$ plume in Fig. 5 and fits for the Birmingham $NO_2$ plume (not shown).

The spatial scale of the a priori makes the greatest difference to the maximum value of the line density, causing a significant

increase in $a$ when the spatial resolution of the a priori profiles increases from 108 km to 12 km. This reflects the impact of the blurring of urban and rural profiles described in Russell et al. (2011).

Both the spatial and temporal resolution impact the determination of $x_0$, the distance traveled in one lifetime. This parameter is determined at fast wind speeds (Lu et al., 2015; Valin et al., 2013), so we consider only the results for wind speed $\geq 3.0$ m s$^{-1}$. For Atlanta, using a daily a priori results in an $x_0$ value 30% greater than that obtained using a monthly average profile at

the same spatial resolution (12 km). The same comparison for Birmingham shows a 66% increase in $x_0$ between the monthly and daily 12 km a priori.

$\mu_x$ represents the apparent center of the $NO_2$ plume relative to the geographic center of the city. This moves downwind (positive) when changing from the monthly average 12 km or 108 km a priori to the daily 12 km a priori. This reflects the ability of the daily a priori to capture how the wind distorts the plume shape.

$\sigma_x$ is the Gaussian smoothing length scale, representing both the width of the upwind Gaussian plume and smoothing of the $NO_2$ signal due to the physical extent of the source, the averaging of $NO_2$ within one OMI pixel, and daily variability in the overpass track (Beirle et al., 2011). There is a slight decrease when going from a monthly average to daily profiles, which reflects the general increase in upwind AMFs (i.e. compare Fig. 1a and 1b), but because this is outside of the main $NO_2$ plume, the effect is small.

Finally, $B$ is the background line density. Ideally, it is derived sufficiently far from any $NO_x$ sources that spatial and temporal variability should be minimal. Indeed, in both fast wind cases no change is observed. When considering slow winds there is a $\sim 25\%$ increase when improving the spatial resolution of the a priori profiles. This is likely attributable to the general increase in urban signal discussed several times so far pulling the edges of the line density upward. However, a greater selection of cities is necessary to demonstrate this more conclusively.

Ultimately, the goal of this method is to extract information about chemically relevant quantities such as emission rate and lifetime. Since de Foy et al. (2014) and Valin et al. (2013) showed that choice of wind speed bins affects the values obtained, we also consider if the effect of implementing the daily a priori profile changes if the observations are binned by different wind speed criteria. Table 3 compares the values of the $NO_x$ emission rate, $E$, and effective lifetime, $\tau_{\text{eff}}$, derived from different wind speed bins for Atlanta and Birmingham. Restricting the analysis to days with wind speed greater than $5 \text{ m s}^{-1}$ results in too few days for a meaningful analysis around Atlanta (due to the need to remove days with winds to the southeast), so results for Atlanta are restricted to $\geq 3 \text{ m s}^{-1}$ and $\geq 4 \text{ m s}^{-1}$ only.

$\tau_{\text{eff}}$ and $E$ are each computed from several of the EMG fitting parameters. $\tau_{\text{eff}}$ depends on $x_0$ and $w$ (the mean wind speed) through Eq. (11):

$$\tau_{\text{eff}} = \frac{x_0}{w} \tag{11}$$

$E$ depends on $a$, $x_0$, and $w$ through Eq. (12):

$$E = 1.32 \times \frac{a \times w}{x_0} = 1.32 \times \frac{a}{\tau_{\text{eff}}} \tag{12}$$

where the factor of 1.32 accounts for the $NO_x$:$NO_2$ ratio throughout the tropospheric column (Beirle et al., 2011).

Both Valin et al. (2013) and de Foy et al. (2014) show that lifetime should decrease at faster wind speeds. We see this trend for Birmingham but not Atlanta. de Foy et al. (2014) also saw that, for a chemical lifetime of 1 h, greater derived emissions were found at faster wind speeds. This is also better seen in our results for Birmingham than Atlanta. Previous measurements of $NO_x$ lifetime in urban plumes average 3.8 h and range from 2-6 h (Beirle et al., 2011; Ialongo et al., 2014; Nunnermacker et al., 1998; Spicer, 1982), and, using the EMG method, Lu et al. (2015) saw effective lifetimes between 1.2 and 6.8 h. The lifetimes we calculate are at the low end of the previously observed ranges. However, this is similar to the instantaneous lifetime of $1.2 \pm 0.5$ h and $0.8 \pm 0.4$ h calculated from the WRF-Chem model for days in June 2013 with wind speed $\geq 3 \text{ m s}^{-1}$ and grid cells within 50 km of Atlanta and Birmingham, respectively (see the supplement for the calculation details).

The differences in the lifetimes and emissions derived using the daily and monthly 12 km a priori profiles are systematic. In all cases, the lifetime derived using the daily profiles is 30–50% longer. When using monthly average a priori profiles, profiles resulting from different wind directions are averaged together. The AMFs calculated from these profiles thus reflect the average distance from the city the plume reaches in a given direction, e.g. east of the city, with smaller AMFs near the city and greater AMFs more distant (Fig. 1). In this hypothetical example, when the wind blows to the east, the spatial extent of

the plume is underestimated because the average AMFs towards the end of the plume will be too large, so the VCDs will be too small by Eq. (1). On days when the wind does not blow east, the reverse is true: the plume extent is overestimated because the AMFs nearer to the city are too small (Fig. 1d). If one considers a simple average change in the VCDs, these two errors will partially cancel and we will see the average change from Sect. 3.2. However, in the EMG fitting approach, these errors do not cancel at all because the EMG method both rotates the $NO_2$ plumes so that the wind directions align before calculating the line densities and systematically selects fast winds to determine $\tau_{\text{eff}}$, so we are always dealing with the first case and the

plume extent is always underestimated. In the EMG fit, this manifests as a too short lifetime. As the emissions are inversely proportional to lifetime (Eq. 12), emissions derived using the monthly 12 km a priori profiles will be too great. Therefore, when using a retrieval with a priori profile at fine spatial resolution, daily temporal resolution of the a priori profiles is necessary to prevent underestimating the lifetime. Further, the spatial resolution of the a priori profiles has a large impact on the magnitude of the derived emissions. To reduce the systematic biases in emissions and lifetime from the choice of a priori profile, it is

necessary to simulate these profiles at fine spatial and daily temporal resolution.

We also use 2-sample $t$-tests at the 95% confidence level (Harris, 2010) to determine if differences in emissions and lifetimes given in Table 3 are significantly different among the results derived from using the three different a priori profile sets for a given city and wind speed bin (i.e. we compare the three values of emissions derived using different a priori profiles for Atlanta and wind speeds $\geq 3$ m s$^{-1}$). This found that, for emissions, the choice of a priori leads to statistically different

emissions for all five cases. For the derived lifetimes, in all cases the monthly 108 km and daily 12 km a priori are statistically indistinguishable, but the monthly 12 km a priori is statistically different. We note that a Durbin-Watson test indicates some spatial autocorrelation remains, and so the uncertainty may be underestimated and the $t$-tests may be incorrectly identifying the differences as significant in this case (Chatterjee and Hadi, 2012). Even if this is true, with a longer averaging period such as those in Beirle et al. (2011), Valin et al. (2013), and Lu et al. (2015), we would expect the random uncertainties to reduce

while the systematic difference from the choice of a priori profile remains. Therefore, the choice of a priori profiles does have an important effect on derived emissions and lifetimes.

We also compare the derived emissions rates to the emissions in a 12 km WRF-Chem model driven by the NEI 11 emission inventory with $NO_x$ emissions scaled to 88.9% of the 2011 values to account for the decrease between 2011 and 2013 (EPA, 2016). WRF-Chem emissions are calculated as the sum of all grid cells within a 50 km radius of the city. 50 km was chosen as

the line densities were integrated for $\sim 50$ km to either side perpendicular to the wind direction. The coarse monthly a priori are 43–61% lower than the NEI-driven emissions, while emissions derived using daily 12 km a priori are within 5–24% (both greater and less than the NEI emissions). Recent work (e.g. Travis et al. 2016 and references within) suggests that the NEI inventory is overestimated by $\sim 50\%$ using both satellite and in situ observations. Emissions derived using daily 12 km show the best agreement to the current NEI inventory, and emissions derived using monthly 108 km a priori profile agree with the

NEI inventory reduced by $50\%$. Therefore, we cannot say which a priori profiles provide the best measurement of emissions by comparing to NEI. It is likely that emissions derived using the monthly 12 km a priori profiles are an overestimate, because the systematically low lifetimes discussed above increase $E$ through Eq. (12); that these emissions are consistently higher than the NEI emission reinforces this likelihood. Conversely, we expect that emissions derived using the coarse monthly a priori

profiles are biased low due to the known underestimate of urban $NO_x$ signals using coarse a priori (Russell et al., 2011). From this, it is clear the choice of a priori profiles has a substantial impact on emissions derived from satellite observations, and that both spatial and temporal resolution of the a priori profiles contribute to that difference. This explains why the OMI derived emissions from Lu et al. (2015) are lower that the bottom up NEI inventory, but needs to be reconciled with work by Travis et al. (2016) which indicates that NEI is overestimated.

In summary, the two most important parameters ($a$ and $x_0$) and values derived from them ($E$, $\tau_{\text{eff}}$) are significantly affected by the spatial and temporal resolution of the a priori. $a$ is most affected by increasing the spatial resolution of the a priori, while using daily profiles corrects a systematic bias in $x_0$ when the profiles are simulated at high spatial resolution. $E$ is affected by both the spatial and temporal resolution of the a priori profiles, increasing by $\sim 100\%$ between the retrievals using coarse monthly and fine daily a priori profiles. Therefore the use of daily a priori $NO_2$ profiles at high spatial resolution significantly alters the results obtained from fitting wind aligned retrieved $NO_2$ columns with an analytical function.

## 5 Conclusions

We have demonstrated that incorporating daily $NO_2$ a priori profiles simulated at sufficiently fine spatial scales to capture the spatial variation of an $NO_2$ plume leads to significant changes in the final VCDs when compared to monthly average profiles at the same spatial resolution. Changes to VCDs on a single day are up to 50% (relative) and $4 \times 10^{15}$ molec. $cm^{-2}$ (absolute). This is attributable to changes in the direction of the $NO_2$ plume. Up to 59% of days with valid observations exhibit changes in VCDs $> 1 \times 10^{15}$ molec. $cm^{-2}$ in at least one pixel. Additionally, the inclusion of daily profiles affects a systematic change in time-averaged VCDs around Atlanta, GA, USA. Pixels downwind in the average exhibited VCD decreases up to 8% ($4 \times 10^{14}$ molec. $cm^{-2}$). Larger relative changes of as much as $-13\%$ were found around the nearby cities of Birmingham, AL and Montgomery, AL. Day-to-day variations in the free troposphere have a smaller impact on the value of the AMF, and average out to no net change over the period studied. These results were obtained using WRF-Chem without lightning $NO_x$ emissions; it is likely that the inclusion of lightning $NO_x$ would increase the magnitude of positive changes to the AMF due to the presence of $NO_2$ at altitude to which OMI is highly sensitive.

When the methods of Lu et al. (2015) are applied to these prototype retrievals, significant changes in derived $NO_x$ emissions are found, increasing by as much as 100% for Atlanta compared to emissions derived from a retrieval using coarse a priori profiles. Using high spatial resolution, monthly average a priori profiles results in the highest derived emissions rates, followed by high spatial resolution, daily a priori, with spatially coarse a priori leading to the lowest derived emissions. Emissions derived using the fine daily a priori are within 25% of the bottom up number from the NEI inventory, a smaller reduction than that suggested by Travis et al. (2016). Future work will aim to resolve this difference. Lifetimes derived from satellite observations using a spatially fine but monthly averaged a priori are systematically biased low due to the spatial pattern of AMF imposed by such a priori; consequently, emissions derived using these a priori profiles are likely biased high. The use of daily profiles at fine spatial resolution corrects this systematic bias.

Having shown that the use of daily a priori $NO_2$ profiles in the retrieval algorithm significantly alters emissions and lifetimes derived from this retrieval, we plan to implement such profiles for several years at the beginning and current end of the OMI data record to investigate how $NO_x$ lifetimes have changed in urban plumes over the past decade. Such work can provide a greater understanding of the most effective means of improving air quality in years to come, as it will allow us to determine whether reductions in $NO_x$ or VOC emissions will provide the most benefit in ozone reduction.

*Acknowledgements.* The authors gratefully acknowledge support from the NASA ESS Fellowship NNX14AK89H, NASA grants NNX15AE37G and NNX14AH04G, and the TEMPO project grant SV3-83019. The MODIS Aqua L2 Clouds 5-Min Swath 1km and 5 km (MYD06_L2) and MODIS Terra+Aqua Albedo 16-Day L3 Global 0.05Deg CMG V005 were acquired from the Level-1 and Atmospheric Archive and Distribution System (LAADS) Distributed Active Archive Center (DAAC), located in the Goddard Space Flight Center in Greenbelt, Maryland (https://ladsweb.nascom.nasa.gov/). We acknowledge use of the WRF-Chem preprocessor tool mozbc, fire_emiss, etc. provided by the

Atmospheric Chemistry Observations and Modeling Lab (ACOM) of NCAR. This research used the Savio computational cluster resource provided by the Berkeley Research Computing program at the University of California, Berkeley (supported by the UC Berkeley Chancellor, Vice Chancellor of Research, and Office of the CIO).

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

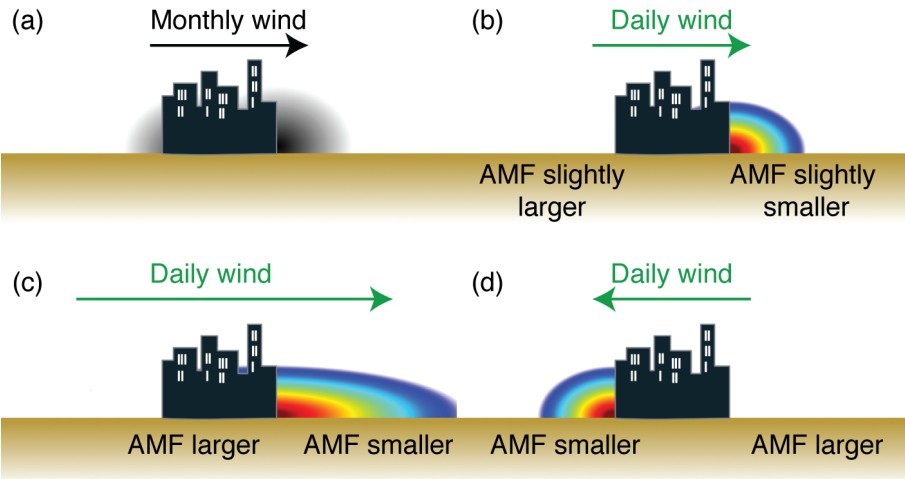

**Figure 1.** An illustration of the central issues that will be discussed in this paper. (a) The monthly average a priori profiles, shown as the grayscale plumes. (b) A case when the daily wind is similar to the monthly average wind. (c) A case where the daily wind is significantly faster than average, but blows in the same direction. (d) A case where the daily wind direction is different than average. The text below each panel describes how the AMF derived from the daily profile would compare with those derived from the monthly a priori.

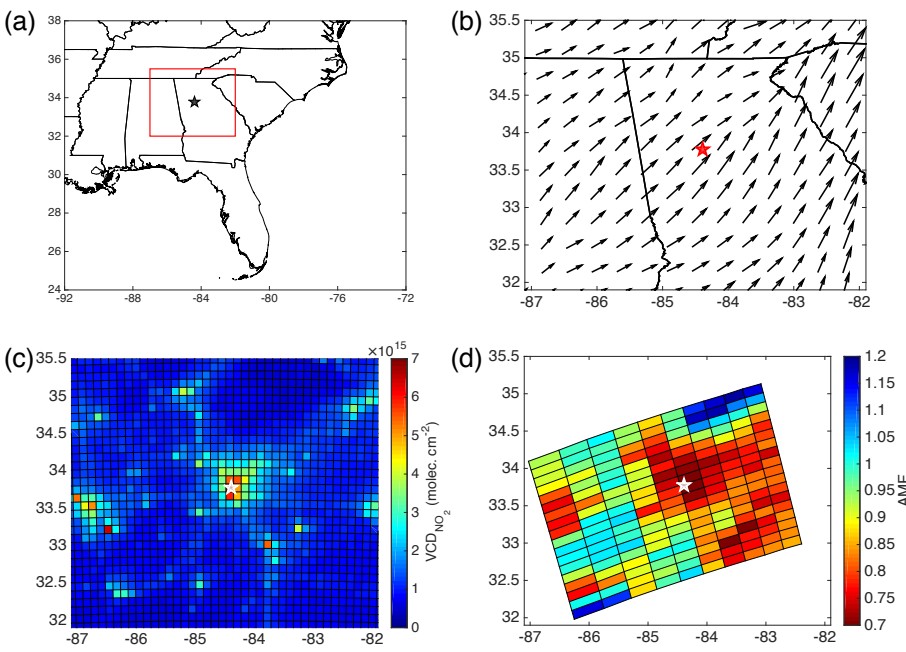

**Figure 2.** Average conditions for June 2013. (a) The red box indicates the part of the SE US being considered. (b) Surface wind directions from the WRF model; average wind speed is 5.0 m s$^{-1}$ (min 1.7 m s$^{-1}$, max 12.7 m s$^{-1}$). (c) WRF-Chem tropospheric NO$_2$ columns. (d) AMFs for the pseudo-retrieval calculated using the average monthly NO$_2$ a priori. The direction of the colorbar is reversed in (d), as small AMFs correspond to high modeled VCDs. In all panels, the star (★) indicates the position of Atlanta. Longitude and latitude are marked on the $x$- and $y$- axes, respectively.

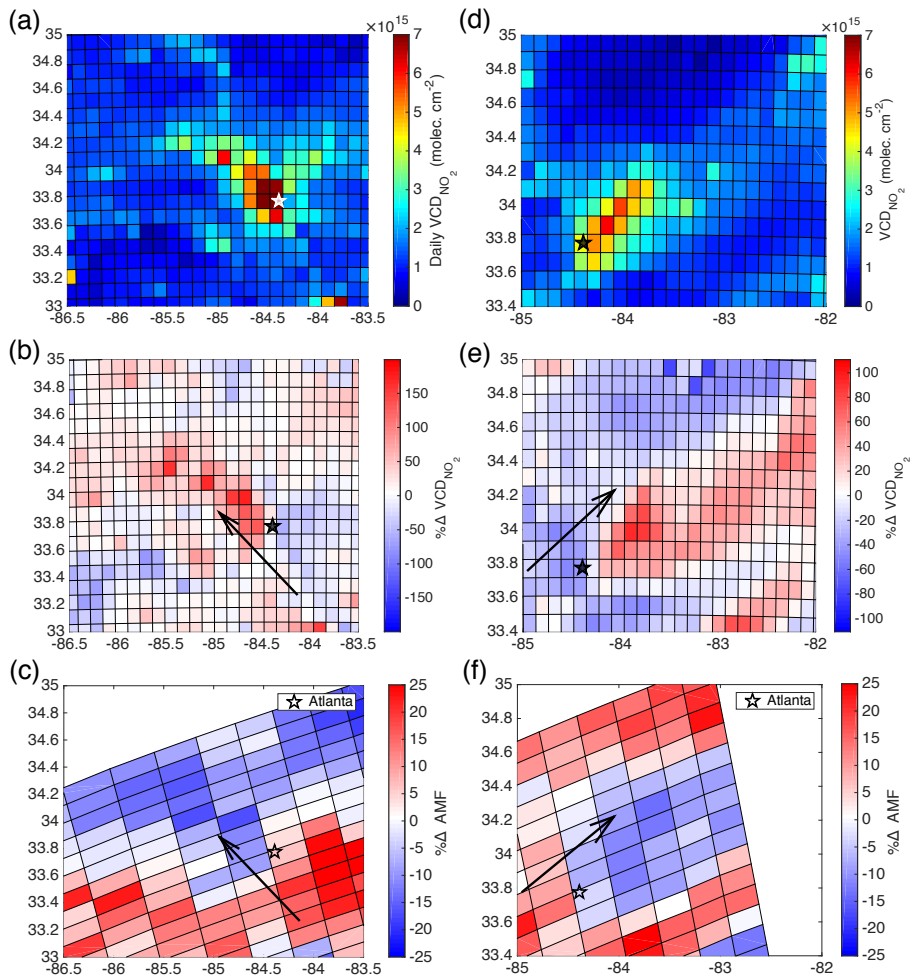

**Figure 3.** Results from 22 June (a–c) and 18 June (d–f). (a,d) WRF-Chem tropospheric $NO_2$ columns for 1900 UTC. (b,e) The percent difference in WRF-Chem tropospheric $NO_2$ columns at 1900 UTC for that day vs. the monthly average. (c,f) Percent difference in AMFs using hybrid daily profiles vs. the monthly average profiles in the pseudo-retrieval. In all panels, the star (★) indicates the position of Atlanta, and the wind direction around Atlanta is shown by the arrow in the lower four panels. Longitude and latitude are marked on the $x$- and $y$-axes, respectively.

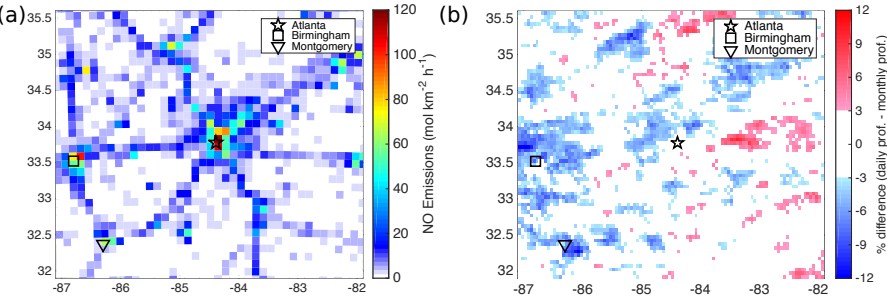

**Figure 4.** (a) 24 h average NO emissions from WRF-Chem at 12 km resolution. (b) The change in retrieved VCDs averaged over 1 June to 30 Aug. Pixels with a cloud fraction $> 20\%$ or that are affected by the row anomaly are excluded from the average. The color scale is reversed from Fig. 3c,f to reflect the inverse relationship between VCD and AMF. Longitude and latitude are marked on the $x$- and $y$- axes, respectively, for both panels.

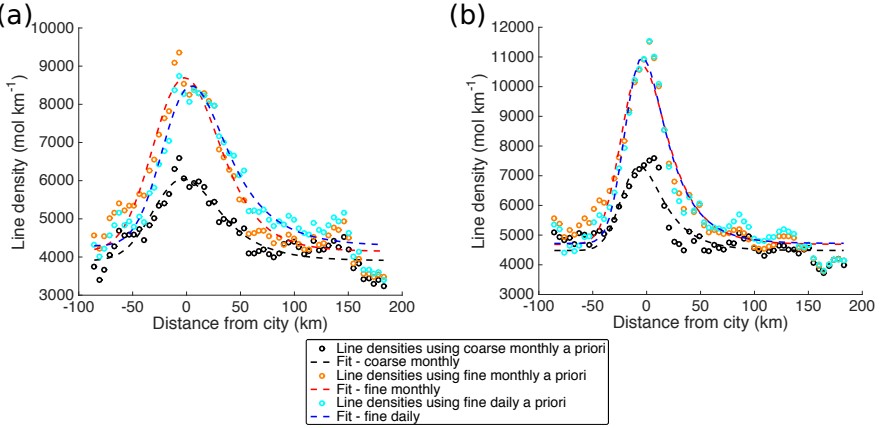

**Figure 5.** Line densities around Atlanta, GA, USA averaged over the study period when using monthly average and daily a priori (open circles), and the corresponding fits of exponentially-modified Gaussian functions (dashed lines). Black series are derived from a retrieval using a monthly average a priori at 108 km resolution; red series from a monthly average a priori at 12 km resolution, and blue from the daily profiles at 12 km resolution. (a) Average of days with wind speed $\geq 3.0$ m/s. (b) Average of days with wind speed $< 3.0$ m/s.

| | Percent of days with $\Delta\text{VCD} > 1 \times 10^{15}$ molec. cm$^{-2}$ | Percent of days with $\Delta\text{VCD} > \left[\sum_i \sigma_i\right]^{1/2}$ | Min. change (molec. cm$^{-2}$) | Max. change (molec. cm$^{-2}$) |
|---|---|---|---|---|
| Atlanta | 39% | 23% | $-2.4 \times 10^{15}$ | $+2.5 \times 10^{15}$ |
| Birmingham | 54% | 43% | $-3.8 \times 10^{15}$ | $+3.9 \times 10^{15}$ |
| Montgomery | 27% | 20% | $-2.2 \times 10^{15}$ | $+1.9 \times 10^{15}$ |

**Table 1.** Statistics on the frequency and magnitude of changes in the retrieved VCDs using a daily vs. monthly average profile for pixels with centers within 50 km of Atlanta, GA, USA (84.39° W, 33.775° N), Birmingham, AL, USA (86.80° W, 33.52° N) and Montgomery, AL, USA (86.30° W, 32.37° N). The "percent of days" values are calculated as the number of days with at least one pixel in that subset with a change greater than the given uncertainty divided by the number of days with at least one pixel unobscured by clouds or the row anomaly. The uncertainty represented by $\left[\sum_i \sigma_i\right]^{1/2}$ is the quadrature sum of uncertainties from spectral fitting ($0.7 \times 10^{15}$ molec. cm$^{-2}$, Boersma et al. 2007, 2011), stratospheric separation ($0.2 \times 10^{15}$ molec. cm$^{-2}$, Bucsela et al. 2013), and AMF calculation (20%, Bucsela et al. 2013).

| | | Wind $\geq$ 3.0 m/s | | | Wind $<$ 3.0 m/s | | |
|---|---|---|---|---|---|---|---|
| | | Monthly | Monthly | Daily | Monthly | Monthly | Daily |
| | | 108 km | 12 km | 12 km | 108 km | 12 km | 12 km |
| Atlanta | $a$ (mol NO$_2$) | $1.8 \pm 0.7 \times 10^5$ | $4. \pm 2 \times 10^5$ | $3. \pm 1 \times 10^5$ | $1.5 \pm 0.6 \times 10^5$ | $4. \pm 1 \times 10^5$ | $3. \pm 1 \times 10^5$ |
| | $x_0$ (km) | $32 \pm 13$ | $26 \pm 11$ | $33 \pm 14$ | $23 \pm 10$ | $24 \pm 10$ | $24 \pm 10$ |
| | $\mu_x$ (km) | $-22. \pm 9$ | $-20. \pm 8$ | $-15. \pm 6$ | $-20. \pm 8$ | $-18. \pm 7$ | $-17. \pm 7$ |
| | $\sigma_x$ (km) | $23. \pm 9$ | $30 \pm 10$ | $22. \pm 9$ | $14. \pm 6$ | $15. \pm 6$ | $13. \pm 5$ |
| | $B$ (mol NO$_2$ km$^{-1}$) | $4. \pm 2 \times 10^3$ | $4. \pm 2 \times 10^3$ | $4. \pm 2 \times 10^3$ | $4. \pm 2 \times 10^3$ | $5. \pm 2 \times 10^3$ | $5. \pm 2 \times 10^3$ |
| Birmingham | $a$ (mol NO$_2$) | $1.6 \pm 0.5 \times 10^5$ | $3. \pm 1 \times 10^5$ | $4. \pm 1 \times 10^5$ | $4. \pm 1 \times 10^5$ | $3. \pm 1 \times 10^5$ | $4. \pm 1 \times 10^5$ |
| | $x_0$ (km) | $50 \pm 20$ | $30 \pm 10$ | $50 \pm 20$ | $220 \pm 80$ | $40 \pm 10$ | $70 \pm 20$ |
| | $\mu_x$ (km) | $-21. \pm 7$ | $-24. \pm 8$ | $-14. \pm 5$ | $-50 \pm 20$ | $-27. \pm 9$ | $-26. \pm 9$ |
| | $\sigma_x$ (km) | $24. \pm 8$ | $27. \pm 9$ | $23. \pm 8$ | $21. \pm 7$ | $25. \pm 8$ | $22. \pm 8$ |
| | $B$ (mol NO$_2$ km$^{-1}$) | $4. \pm 1 \times 10^3$ | $4. \pm 1 \times 10^3$ | $4. \pm 1 \times 10^3$ | $4. \pm 1 \times 10^3$ | $5. \pm 2 \times 10^3$ | $5. \pm 2 \times 10^3$ |

**Table 2.** Values of the five fitting parameters for the EMG functions (Eq. 9) used to fit the distributions of line densities around Atlanta and Birmingham. $a$ represents the total NO$_x$ burden, $x_0$ is the distance the plume travels in one lifetime, $\mu_x$ is the center of emissions relative to the city center, $\sigma_x$ describes the Gaussian smoothing, and $B$ the background line density.

|  | Wind speed bin | Atlanta | | | Birmingham | | |
|---|---|---|---|---|---|---|---|
|  |  | Monthly 108 km | Monthly 12 km | Daily 12 km | Monthly 108 km | Monthly 12 km | Daily 12 km |
| E (Mg NO$_x$ h$^{-1}$) | WRF-Chem NEI |  | **13.74** |  |  | **10.49** |  |
|  | $\geq 3.0$ | $6. \pm 4$ | $16. \pm 9$ | $11. \pm 7$ | $4. \pm 2$ | $10. \pm 6$ | $8. \pm 5$ |
|  | $\geq 4.0$ | $6. \pm 3$ | $17 \pm 11$ | $11. \pm 6$ | $4. \pm 2$ | $13. \pm 7$ | $9. \pm 5$ |
|  | $\geq 5.0$ | - | - | - | $6. \pm 3$ | $15. \pm 9$ | $11. \pm 6$ |
| $\tau$ (h) | $\geq 3.0$ | $1.6 \pm 0.7$ | $1.3 \pm 0.5$ | $1.7 \pm 0.7$ | $2.5 \pm 1.0$ | $1.8 \pm 0.7$ | $2.6 \pm 1.0$ |
|  | $\geq 4.0$ | $1.8 \pm 0.7$ | $1.2 \pm 0.5$ | $1.8 \pm 0.7$ | $2.1 \pm 0.9$ | $1.5 \pm 0.6$ | $2.2 \pm 0.9$ |
|  | $\geq 5.0$ | - | - | - | $1.8 \pm 0.7$ | $1.3 \pm 0.5$ | $1.8 \pm 0.7$ |

**Table 3.** Values of the emission rates ($E$) and effective lifetime ($\tau$) obtained when the separation between slow and fast winds is set at 3, 4, and 5 m s$^{-1}$. For comparison, the total NO$_x$ emission for all 12 km WRF-Chem grid cells within 50 km of each city is given. These emissions are derived from NEI 11 and scaled to 88.9% to account for 2011–2013 reductions. Uncertainties calculated as described in the supplement.