# Peer review of "Effects of daily meteorology on the interpretation of space-based remote sensing of $NO_{\rm 2}$"

_Atmospheric Chemistry and Physics, 2016_

## Referee Comment (RC1) · Anonymous Referee #1 · 9 Aug 2016

The manuscript entitled "Effects of daily meteorology on the interpretation of space-based remote sensing of NO2", by Laughner et al., demonstrates the importance of high temporal and spatially generated airmass factors when calculating OMI tropospheric column NO2. From their study, the apriori used for the retrieval comes from the WRF-Chem model and the temporal and spatial resolutions are investigated when calculating the tropospheric column NO2.

Overall, this is a nice study and highlights the some of the limitations in the current publically available NO2 products (i.e. NASA and KNMI). For that reason, the manuscript should be published in ACP with some minor corrections.

Minor comments:

Figure 1 is a very nice example of how the AMF is influenced by monthly and daily aprioris. From the manuscript, the authors suggest that this is based on the prior vertical information of NO2 from the model and the scattering weighting function. However, in the introduction there was no reference to why the AMFs are smaller/larger in the scenarios in Figure 1. I think adding a short explanation of what causes the AMFs to be different would be useful. Also, as far as I can tell, the manuscript doesn't quantify which of the two processes in Eqn 2 & 3 influencing the AMFs are most important. Which is it?

In section 2.5 the EMG is discussed in detail (and in the supplementary information), but often with reference to how Matlab functions are used to calculate the required equations. For someone who has not used Matlab before, this might be difficult to follow (e.g. fmincon, nansum etc). Could the authors just discuss the mathematical and statistical methods used and leave discussion of Matlab functions in the supplementary material? I also think that there should be discussion on what a, x0, $\mu$x, $\sigma$x and B are in the text of section 2.5, instead of just referring to Table 1.

In section 2.3, there is discussion on weighting schemes (i.e. Eqn 5). Unfortunately, I do not understand how and why this is used. If you are trying to calculate the model monthly mean relative to the OMI sampling, could you not just subsample the model to the individual satellite overpass times (e.g. within 1 hour of 14.00LT)? This text (Page 6, Lines 5-10) needs to be improved to make the motivation for Eqn 5 clearer.

In the abstract and introduction any reference to "Atlanta, GA" should be "Atlanta, GA, USA" as not everyone will know Atlanta is in the USA.

On Page 2, Lines 26-27, "Because the satellite is less sensitive to NO2 near the surface, the AMF should be smaller in locations influenced by surface NOx sources" this is discussed in detail throughout the manuscript, but it would be good to add a short sentence here explaining why this is the case.

Page 3, Line 11: BEHR needs to be defined here, not later in the section.

Page 3, Lines 13-14: "2° x 3° and 0.5° x 0.667°". I'm assuming this is lons then lats?

For the statement "Valin et al. (2013) showed that the concentration of NO2 downwind of a city increases significantly with wind speed." on Page 3 Lines 24-25, can a range be provided to which this statement is true.

Page 3, Lines 27-29: The authors should explain why OMI is less sensitive at lower altitudes or provide a reference which explains this.

In the Introduction, we are informed that this study focuses on Atlanta. The reasoning for this is explained later on in the manuscript, but needs to be mentioned here as well to make it clear why this region is the focus of the study.

Please reword Lines 30 (P4) – 2 (P5) to make the text clearer. i.e. "These have been classified as the "row anomaly" and as of 5 July 2011 affect one-third of the pixels http://projects.knmi.nl/omi/research/product/rowanomaly-background.php), reducing coverage from global daily to global every two days".

In the title of section 2.2 please expand out what "BEHR" represents. Also, P5, L5, BEHR doesn't need to be expanded again as it is done so in the Introduction.

In Eqn 2,3 please specifically state what "p" represents. Also, how is ptp defined (e.g. dynamical, chemical tropopause)?

Please provide a reference or explanation why "an assumed cloud albedo of 0.8" is used, on P5 , L 21.

P5 Line 29: What does NE1 11 stand for/represent?

The manuscript should not reference papers in prep such as Zare, (in prep) on P6 Line 4.

P6, L5: Please provide justification for using a spin up period of just 5 days.

P6, L20: Please rephrase ""pseudo-retrieval" that is much simplified compared to a full

operational NO2 retrieval" and outline how it is much simpler. Also on L21, please state some examples of co-founding variables.

Section 2.5: Firstly, I cannot find an example of where EMG is expanded (i.e. what does it stand for)? Secondly it should be expanded in the introduction where EMG is first mentioned. It should also be written in full for the title of this section.

P7 L20: An explanation on why the WRF winds are transformed to earth-relative from grid-relative would be much appreciated.

Eqn 8: Please define specifically what F(x. . ...) represents. Also stating what "erfc" stands for would be useful as well. I'm assuming it is "error function"?

P9, L8-9: As stated above, the reasoning for choosing Atlanta needs to be outlined earlier on in the manuscript.

In Figure 2b, it would be useful to add a scale for the wind speed or add some text to the caption stating what the min, mean and max winds in the domain are for that period.

P9-10, L30-1: Please expand on "All pixels show a positive change." Is this correct. Should there not be negative changes somewhere in the domain?

Please expand VCDs in the section 3.2 title.

P11, L9-11, the authors state that the uncertainty value of 1015 molecules per cm2 can be reduced by a factor of the SQRT(n). However, this assumes that all errors in this uncertainty value are random. Surely, some of the error will be systematic or smoothing errors? Therefore, the authors show take this into account or explain why it can be done like this.

The two sentences on P11, L7-9 "The main decrease around Atlanta is to the northeast, along the direction that the monthly average model results placed the NO2 plume. A systematic decrease of 5–10% to the northeast of Atlanta is observed; this is the

plume direction in the monthly average profiles." need to be reworded as discussing "northeast" twice is repetitive.

Just double checking on P11, L17, this should definitely "southeast"?

P12, L12. Should be "a x0" and not "an x0".

On P13, L14-19, the t-test is discussed to determine if "differences in emissions and lifetimes are significantly different among the results derived from using the three different a-priori profiles....". The t-test assumes that data within the sample population are independent. However, I imagine there will be lots of temporal autocorrelations in the samples. Do the authors account for this and if not, why?

On P14, Lines 3-15, comparisons to the NEI 11 emissions are discussed. From the text, the NE1 11 emissions are overestimated by 50%. If the NEI 11 emissions took this into account, which results (e.g. monthly or daily apriori ) which have best agreement with them. Also, on L7, the authors state that the "daily 12 km a priori are within 5-24%". Are they lower or higher than the NEI 11 emissions or both? The authors have stated that the "coarse monthly a priori" are lower by 43-62%.

Supplementary Information:

P1, L16: Why would we expect the mean and median differences to be 0?

P1, L20-22: "This, combined with the greater scattering weights at these altitudes, explains why the effect on the AMF is as large as it is, although day-to-day changes in the boundary layer still dominate the effect using daily profiles has on the AMFs,". The "," at the end of the sentence should be a full stop. Secondly, can the impact of the scattering weights at this altitude be quantified, say in comparison to the impact of the apriori on the AMFs?

P1, L30: Is this period of 91 days long enough for the averaging to have no net impact?

P2, L 9: Should be "Atlanta, nevertheless" or "Atlanta. Nevertheless".

[Figure]

P4, L4-5: Should be these "]" brackets and not ")"?

P5, L20: "as the errors contributing to it should be random in nature". Why is this the case? No systematic errors?

P6, L15: "). nfit"?
* * *

---

## Referee Comment (RC2) · Anonymous Referee #2 · 8 Sep 2016

Review of "Effects of daily meteorology on the interpretation of space-based remote sensing of NO2" by J. L. Laughner et al.

In this manuscript, the authors investigate the impact of using daily instead of monthly high resolution a priori NO2 profiles in satellite retrievals of tropospheric NO2 plumes used for emission estimates. They explore qualitatively what the expected changes are, investigate the effects more quantitatively in simulated retrievals and finally apply daily a priori profiles to a limited set of real OMI data.

The manuscript is well written and nicely explains the basic effects and how they impact on the results. The topic of study – improvements of NOx emission estimates from satellite NO2 measurements – is relevant for the community and the relatively large

sensitivity reported is important for the interpretation of existing and future studies. As the paper is more on the technique and less on the impact on atmospheric chemistry and physics I think it would have been more appropriate for AMT but it also fits into the scope of ACP. I therefore recommend it for publication after revisions taking into account my comments listed below.

1. While I see the nice qualitative discussion of the effects as a strong point of this manuscript, the quantitative results are much less convincing in my opinion. The reason for my scepticism is the large change in emissions and lifetimes the authors find when changing their a priori spatial or temporal resolution in combination with the large uncertainties given in the tables. Most of the results are in agreement within uncertainties when changing from monthly to daily a priori in spite of the large changes seen. To me this indicates that the time period used for averaging is too short to really separate the effect of a priori changes from noise in the data, and as the authors acknowledge, all previous studies used much longer averaging times. Please comment on the magnitude of uncertainties and the significance of differences seen.

2. In the discussion of the results both in the text and in the abstract, I'm confused by statements such as "Comparing an optimized retrieval to a more standard one, we find that NOx emissions estimated from space-based remote sensing can increase by 100% when daily variations in plume location and shape are accounted for in the retrieval." If I'm not misinterpreting Table 4, the change in emissions when moving from monthly to daily a priori is closer to 45% and actually is a decrease, not an increase of estimated emissions. This also makes more sense considering the qualitative discussion given in the first part of the paper. The factor of 2 increase is relative to a low spatial resolution a priori which is also interesting but not the focus of the study and also not what is suggested by the formulations in the text. I think these statements need to be rephrased.

3. The discussion of standard mathematical methods for fitting a function to the decay curve is a bit out of place in such an article, in particular as the method used for emission estimation is not the topic of the paper. I would suggest to shorten this part and to remove the discussion of Matlab internals which are of little interest to the reader.

4. The approach taken to averaging the model data in time (Equation 5) appears overly complex and not transparent. The obvious way to treat this problem is simple interpolation in time to the OMI overpass.

5. On a more general note I think that the paper would benefit from a short discussion of the impact model errors could have on the results. It is obvious that in theory, using daily a priori data is better than using monthly averages as the process of NO2 retrieval is not linear. However, in real data this might not necessarily be true. At the high spatial resolution of the model used here, even a small uncertainty in wind direction, emission height or emission source location can move the NO2 plume into different model grid cells, potentially leading to poor matching of plume position in measurement and model and thus wrong air mass factors and NO2 columns. I find this an interesting topic in particular in view of future instruments having improved spatial resolution.

6. Equation 2: $w_s(p)$ is later written as $w(p)$

7. Equation 4: Please add how cloud radiance fraction was computed

8. Page 6: Add reference for MOZART model

9. Page 7, line 5: Cloudy AMF is smaller, not lager for boundary layer NO2 profiles

10. Page 14, line 17: something missing here?

11. Conclusions, line 28: Please add again that you count days even if only a single pixel shows a change larger than the noise (which I personally find a strange way of counting)

12. Pageline 6: I do not understand the sentence "Further work is needed to understand the impact of this change on top down constraints of NOx emissions, given the recent work showing that bottom up estimates are high by 50%". I think the relevance is obvious if you believe your own results – all previous estimates based on this technique and using monthly high resolution a priori profiles give too high emission estimates. Whether or not the emission inventory is off is another topic (which could of course also impact on your AMF values and thus emission estimates) and should not be mixed here.

---

## Referee Comment (RC3) · Anonymous Referee #3 · 8 Sep 2016

The study by Laughner et al. addresses the relevant issue of highly resolved a priori profiles for NO2 retrievals from OMI. The issue –a priori NO2 profile information is required at the spatial resolution of an OMI pixel ($\sim$300 km2) for the actual day *and* time of the measurement– has been known for a long time (e.g. Boersma et al., 2007; Heckel et al., 2011). The merit of this study is that the authors quantify the effect of diurnal and highly resolved a priori NO2 profiles on the air mass factor calculations, relative to AMFs using either coarse profiles or monthly mean profiles. As expected, considerable differences are found in the retrieved NO2 columns, and these differences can be interpreted as systematic error contributions in operational retrievals making use of coarse spatial resolution or monthly mean averaged NO2 profiles in their AMFs.

[Figure]

Moreover, the authors show that this undersampling problem has serious consequences for the recently developed technique to simultaneously estimate NOx emissions and NO2 lifetime for isolated pollution sources from wind-sorted satellite measurements of tropospheric NO2 columns (Beirle et al., 2011). The authors address those consequences by comparing the effect of using NO2 retrievals with different profiles (high vs. low spatio-temporal resolution) on the final estimates of NOx emissions and lifetimes. In doing so, the authors use an interesting approach: that of a 'pseudo-retrieval'. It allows to more or less investigate the effect of profile changes on the retrievals without the usual perturbing influences from other retrieval parameters such as variable viewing geometries, clouds, albedo, etc. They find much higher NOx emissions from satellite retrievals using highly-resolved a priori profiles, and this is an intriguing result in view of previously published city emission estimates using retrievals with coarser a priori profiles, as the differences are easily a factor of 2.

Major issues:

* One limitation is the focus on only one area (Atlanta, GA) and one season (Summer). As pointed out by another reviewer, focusing on such a short period leads to a limited dataset, from which it is difficult to obtain estimates of emissions and lifetimes that are significantly different. Since the differences in the estimated emissions are at least a factor of 2, it should be feasible to achieve statistically significant differences by analysing a longer period. This would strengthen the paper considerably.

* Then there is a serious error in the theoretical framework for BEHR AMFs. According to Eq. (2) and the text in section 2.2, the cloudy AMF is calculated only between the cloud pressure level and the tropopause. In principle this can be done, but then the retrieval needs to account for a so-called "ghost column" [Burrows et al., 1999]. A ghost column correction however, is not being applied here. The better alternative is to calculate both the clear-sky and cloudy-sky AMFs by integrating the NO2 profile from the surface pressure to the tropopause. This formulation ensures that the AMF value returns a tropospheric column that is representative for all NO2 in the troposphere,

and not just for the above-cloud fraction (in the limiting case of cloud fraction 1.0). The authors should revisit their integration limits for Eq. (2), probably also for their current BEHR-product.

This error is also the reason for the incorrect statement on page 7, under point 3: "... a cloudy pixel will have a much greater AMF than a clear one". The statement would be correct only if the word 'AMF' would be replaced by 'above-cloud AMF'. And, in line with the above criticism, an above-cloud AMF can only return an above-cloud NO2 columns, which cannot be the purpose of a state-of-science retrieval.

* There are difficulties with the interpretation of uncertainties in the tropospheric column retrieval. On page 10, the choice for 1.0x10ˆ15 molec. cmˆ-2 as a typical number for the uncertainty in the tropospheric NO2 column is rather arbitrary. Detailed error analyses (e.g. Boersma et al. [2004]) have pointed out that the uncertainty in the tropospheric column is highly variable because of AMF (a multiplicative factor indeed) uncertainties, which depend strongly on variable clouds, surface albedo, and NO2 profile shapes. Moreover, the estimate used by the authors is rather optimistic. More realistic uncertainties are on the order of 1.0x10ˆ15 molec. cmˆ-2 +25% (of the individual column value).

Also, part of the AMF (25%) uncertainty is random in nature, and part is systematic. So by averaging over a large number of pixels, as is done on page 11, only the random part of the AMF-related uncertainty reduces, but not the systematic part. This makes the estimate of the 'nominal uncertainty' of ∼1.6x10ˆ14 molec.cmˆ-2 too optimistic. The given value may hold for the SCD-related uncertainty, but is not representative for the uncertainty in averaged tropospheric NO2 columns, where systematic (e.g. albedo-related or cloud-related) errors are likely still of concern.

Specific issues:

Abstract, line 6: this paper does not address variations of NO2 in power plant plumes, so this should be removed. The paper is about NO2 variations in urban plumes.

[Figure]

Introduction, lines 18 and 30: a reference to the work by Vinken et al., ACP, 2014 on ship emissions estimates using improved-resolution a priori profiles would be appropriate here.

P3, lines 7-8: earlier studies by McLinden et al. [2012] showed that the oil sand signals was detectable also for retrievals using coarse-gridded a priori profiles. The statement should be nuanced in that the spatial signatures are more realistically resolved with higher resolution a priori profiles.

P3, L13: also include here a reference to the DOMINO retrievals using 0.5 x 0.67 profiles from GEOS-Chem over Europe from Vinken et al. [2014]. These retrievals also use diurnal profiles.

P3, L17: the NASA retrieval is usually indicated as the Standard Product v2 or SP v2. OMNO2 (actually OMNO2A v1) refers to the software for DOAS spectral fitting.

P4, L30: 13 x 24 km2

P4, L32: 'theoretical' daily global coverage is a strange term. Coverage was nearly global every day until the row anomaly, and after that, coverage is more or less global every 2 days.

P5, L6: please provide the name and appropriate version of the NO2 SCD product used. I think it is OMNO2A v1. This product has recently been evaluated in Marchenko et al. [2015] and van Geffen et al. [2015]. It would be appropriate to cite those papers here.

P5, L17: apart from albedo, please also provide details on the cloud information (effective cloud fraction, pressure) used in the BEHR-approach.

P5, L27: which version of the WRF-Chem model is used?

P6, section 2.3: please provide some more details on the WRF-Chem model such as what is the NOx emission total over the US in the period of interest, and on the meteorological and chemical boundary conditions used. It is unclear how realistic WRF-Chem simulations of NO2 profiles are. Have these been validated against surface and aircraft measurements of NO2?

P7, L19: please clarify what the vertical range is of the 'first five layers' of WRF-Chem.

P8, L6: is it really necessary or relevant to refer to a Matlab function, when explaining how you do the fit? If it is, please provide some more information on why you chose this particular fit approach over other alternatives. As a non-Matlab user, the sentence does not mean much to me. Later on the same for 'fmincon'.

P8, 12-17: this part is very technical and should be moved to an appendix or supplement.

P10, L13: that lightning is not included in this WRF-Chem set-up should not be mentioned only here, but already in the model description section 2.3.

P12, L18: sigma_x represents the width of the Gaussian plume, but also the spatial smearing of the signal caused by the satellite pixel extent, and the fact that cities are covered by different satellite footprints from day to day.

P13, L14-19: this whole section presumably discusses Table 4, but that is not obvious from the text. I'm confused by the statement that the "choice of a priori leads to statistically different emissions for all five cases", whereas Table 4 shows emission values that all overlap within the quoted uncertainty estimates.

References

Boersma, K. F., H. J. Eskes, and E. J. Brinksma (2004), Error analysis for tropospheric NO2 retrieval from space, J. Geophys. Res., 109, D04311, doi:10.1029/2003JD003962.

Boersma, K. F., Eskes, H. J., Veefkind, J. P., Brinksma, E. J., van der A, R. J., Sneep, M., van den Oord, G. H. J., Levelt, P. F., Stammes, P., Gleason, J. F., and Bucsela, E.

J.: Near-real time retrieval of tropospheric NO2 from OMI, Atmos. Chem. Phys., 7, 2103-2118, doi:10.5194/acp-7-2103-2007, 2007.

Burrows, J. P., Weber, M., Buchwitz, M., Rozanov, V., Ladstätter-Weißenmayer, A., Richter, A., DeBeek, R., Hoogen, R., Bramstedt, K., Eichmann, K. -U., Eisinger, M., and D. Perner, The Global Ozone Monitoring Experiment (GOME): Mission Concept and First Scientific Results, J. Atmos. Sci., 56:151-175, 1999

Heckel, A., Kim, S.-W., Frost, G. J., Richter, A., Trainer, M., and Burrows, J. P.: Influence of low spatial resolution a priori data on tropospheric NO2 satellite retrievals, Atmos. Meas. Tech., 4, 1805–1820, doi:10.5194/amt-4-1805-2011, 2011.

Marchenko, S., N. A. Krotkov, L. N. Lamsal, E. A. Celarier, W. H. Swartz, and E. J. Bucsela (2015), Revising the slant column density retrieval of nitrogen dioxide observed by the Ozone Monitoring Instrument. J. Geophys. Res. Atmos., 120, 5670–5692. doi: 10.1002/2014JD022913.

McLinden, C. A., V. Fioletov, K. F. Boersma, N. Krotkov, C. E. Sioris, J. P. Veefkind, and K. Yang (2012), Air quality over the Canadian oil sands: A first assessment using satellite observations, Geophys. Res. Lett., 39, L04804, doi:10.1029/2011GL050273.

van Geffen, J. H. G. M., Boersma, K. F., Van Roozendael, M., Hendrick, F., Mahieu, E., De Smedt, I., Sneep, M., and Veefkind, J. P.: Improved spectral fitting of nitrogen dioxide from OMI in the 405–465 nm window, Atmos. Meas. Tech., 8, 1685-1699, doi:10.5194/amt-8-1685-2015, 2015.

Vinken, G. C. M., Boersma, K. F., van Donkelaar, A., and Zhang, L.: Constraints on ship NOx emissions in Europe using GEOS-Chem and OMI satellite NO2 observations, Atmos. Chem. Phys., 14, 1353-1369, doi:10.5194/acp-14-1353-2014, 2014.

---

## Author Comment (AC1) · 21 Oct 2016

**Effects of daily meteorology on the interpretation of space-based remote sensing of NO$_2$**

**Response to Anonymous Referee #1**

Joshua L. Laughner, Azimeh Zare, and Ronald C. Cohen

October 20, 2016

We thank the reviewer for their positive response and very careful reading of both the main article and the supplement. Below we respond to the individual comments. The reviewer's comments will be shown in red, our response in blue, and changes made to the paper are shown in black block quotes. Unless otherwise indicated, page and line numbers correspond to the original paper. Figures, tables, or equations referenced as "R$n$" are numbered within this response; if these are used in the changes to the paper, they will be replaced with the proper number in the final paper. Figures, tables, and equations numbered normally refer to the numbers in the original discussion paper.

Figure 1 is a very nice example of how the AMF is influenced by monthly and daily a prioris. From the manuscript, the authors suggest that this is based on the prior vertical information of NO2 from the model and the scattering weighting function. However, in the introduction there was no reference to why the AMFs are smaller/larger in the scenarios in Figure 1. I think adding a short explanation of what causes the AMFs to be different would be useful. Also, as far as I can tell, the manuscript doesnt quantify which of the two processes in Eqn 2 & 3 influencing the AMFs are most important. Which is it?

The paragraph on P4, L7–18 discusses each of the scenarios in Fig. 1. We have added a sentence near the beginning of this paragraph to remind the reader of the physical cause for the dependence on the NO$_2$ profile illustrated in each panel, so the paragraph now begins as:

> "In this paper we explore how day-to-day changes in the a priori NO$_2$ profiles affect satellite retrievals of urban NO$_2$. Several scenarios are illustrated in Fig. 1. In each case the change in the AMF results because, over low albedo surfaces, a UV/visible satellite spectrometer is less sensitive to near surface trace gases, necessitating a smaller AMF to account for the reduced sensitivity. In Fig. 1a..."

Regarding which process is more important, and taking this to mean the calculation of scattering weights vs. the a priori profile, there is no one answer, as this varies with the amount of NO$_2$ present. We have expanded the part at P.1 L.25–27 to explain this and noted that previous work has found a priori profiles to be very significant near urban areas:

"...the AMF must account for the varying sensitivity of the satellite to $NO_2$ at different altitudes, and therefore a priori knowledge of that sensitivity and the vertical profile of $NO_2$ is required. Over low-reflectivity surfaces, light scattered in the atmosphere is the primary source of radiance at the detector. The probability of back-scattered light penetrating to a given altitude is greater for higher altitudes; thus there is greater interaction with, and therefore greater sensitivity to, $NO_2$ at higher altitudes (Richter and Wagner, 2011; Hudson et al., 1995). Because of this, the correct AMF is smaller in locations influenced by surface $NO_x$ sources. The relative contribution of errors in the calculated sensitivity and in the a priori profiles of $NO_2$ to error in the final VCD varies between polluted and clean pixels (Boersma et al., 2004). Previous work (e.g. Russell et al. 2011) has sought to reduce errors in both, and highlighted the importance of accurate a priori profiles in urban areas."

In section 2.5 the EMG is discussed in detail (and in the supplementary information), but often with reference to how Matlab functions are used to calculate the required equations. For someone who has not used Matlab before, this might be difficult to follow (e.g. fmincon, nansum etc). Could the authors just discuss the mathematical and statistical methods used and leave discussion of Matlab functions in the supplementary material? I also think that there should be discussion on what a, x0, $\mu$x, $\sigma$x and B are in the text of section 2.5, instead of just referring to Table 1.

This comment is well taken. Similar comments were made by other reviewers. We have moved the technical elements (P.8 L.9–29) to the supplement and included a description of the fitting parameters here instead, after Eq. (9):

"Eq. (9) is minimized using an interior-point algorithm, finding the values of $a$, $x_0$, $\mu_x$, $\sigma_x$, and $B$ that best fit the line densities. The values of $a$, $x_0$, $\mu_x$, $\sigma_x$, and $B$ have physical significance and so their optimum values yield information about the $NO_x$ emission and chemistry occurring within the plume (Beirle et al., 2011; de Foy et al., 2014; Lu et al., 2015). Specifically:

- $a$ describes the total amount of $NO_2$ in the plume (referred to as the burden)

- $x_0$ is the distance the plume travels in one lifetime, $\tau$. It relates to $\tau$ by $x_0 = \tau \times w$, where $w$ is wind speed.

- $u_x$ describes the effective center of the emission source. In the supplement to Beirle et al. (2011), it is represented by $X$ which is the point at which exponential decay of the $NO_2$ plume begins.

- $\sigma_x$ is the standard deviation of the Gaussian component of the EMG function. Lu et al. (2015) terms this a "smoothing length scale," which describes smoothing of the data due to the spatial resolution and overlap of OMI pixels (Boersma et al., 2011). It can also be thought of as capturing effects of both the spatial extent of emissions and the turbulent wind field.

- $B$ is the background line density.

"

In section 2.3, there is discussion on weighting schemes (i.e. Eqn 5). Unfortunately, I do not understand how and why this is used. If you are trying to calculate the model monthly mean relative to the OMI sampling, could you not just subsample the model to the individual satellite overpass times (e.g. within 1 hour of 14.00LT)? This text (Page 6, Lines 5-10) needs to be improved to make the motivation for Eqn 5 clearer.

We have added a paragraph after Eq. (5) explaining our reasoning for these weights:

"The weighting scheme in Eq. (5) was chosen over simply using the model output for 1400 local standard time for each longitude to create smooth transitions between adjoining time zones. This attempts to account for the day-to-day variability in OMI overpass tracks as well as the fact that pixels on the edge of a swath can be observed in two consecutive overpasses at different local times. More detail is given in the supplement."

And in the supplement:

"When computing the monthly average profiles, it is necessary to use profiles that represent OMI's overpass time, typically quoted as 13:30 to 13:45 local standard time (e.g. McLinden et al. 2014; Levelt et al. 2006). To average the profiles output from WRF-Chem, weights were calculated that fulfilled two requirements:

1. The weights should be 1 at OMI overpass time and 0 when more than 1 hour away from overpass time.

2. The transition between profiles from different hours should be smooth.

For #1, we assume that the average overpass time is 1330 local standard time. We compute local standard time as:

$$t_{\text{apriori, local}} = \frac{l}{15} + t_{\text{apriori, utc}} \qquad \text{(R1)}$$

where $t_{\text{local}}$ is the local standard time in hours past midnight, $t_{\text{utc}}$ the UTC time in hours past midnight, and $l$ the longitude (west is negative). To meet the second requirement, this is a continuous function, rather than a step function (where each 15° longitudinal segment/time zone has a single local time). Areas further west in a time zone are more likely to be observed on the east edge of a later OMI swath, and vice versa for areas further east. This weighting includes some influence from later profiles to account for this.

The weights from Eq. (5) are derived from:

$$w = 1 - |t_{\text{overpass}} - t_{\text{apriori, local}}| = 1 - \left| 13.5 - \frac{l}{15} - h \right| \qquad \text{(R2)}$$

[Figure]

Figure R1: Swaths covering the east coast of the US for 2 June 2013 (a,b) and 3 June 2013 (c,d). The times given are the start and end times of the daytime half of the orbit in UTC. As shown, on different days, the time of the OMI swath that covers Atlanta can vary by up to an hour.

where $t_{\text{overpass}}$ is the assumed overpass time for OMI and $h \equiv t_{\text{apriori, utc}}$. If $w < 0$, $w$ is set to 0. This gives us the desired form where the weights smoothly vary in time.

"

In the abstract and introduction any reference to "Atlanta, GA should be "Atlanta, GA, USA as not everyone will know Atlanta is in the USA.

We have added "USA" to any instance where the city is given as "Atlanta, GA," and did the same for "Birmingham, AL" and "Montgomery, AL." Cases of the city name alone were left as such (i.e. just "Atlanta" not "Atlanta, GA").

On Page 2, Lines 26-27, "Because the satellite is less sensitive to NO2 near the surface, the AMF should be smaller in locations influenced by surface NOx sources this is discussed in detail throughout the manuscript, but it would be good to add a short sentence here explaining why this is the case.

We have added two sentences before Eq. 1 explaining the physical basis for the lower near-surface sensitivity:

"Over low-reflectivity surfaces, light scattered in the atmosphere is the primary source of radiance at the detector. The probability of back-scattered light penetrating to a given altitude is greater for higher altitudes; thus there is greater

interaction with, and therefore greater sensitivity to, $NO_2$ at higher altitudes (Richter and Wagner, 2011; Hudson et al., 1995). Because of this, the correct AMF is smaller in locations influenced by surface $NO_x$ sources."

Page 3, Line 11: BEHR needs to be defined here, not later in the section.
Definition of BEHR added.

"The current generation Berkeley High Resolution (BEHR) (Russell et al., 2011, 2012) and OMI-EC (McLinden et al., 2014) retrievals simulate monthly average $NO_2$ profiles at 12 and 15 km, respectively."

Page 3, Lines 13-14: "$2°$ x $3°$ and $0.5°$ x $0.667°$. Im assuming this is lons then lats?
These are actually lat x lons. We have reversed the order and clearly defined it:

"Conversely, the DOMINOv2 (Boersma et al., 2011), POMINO (Lin et al., 2015), and DOMINO2_GC (Vinken et al., 2014) retrievals simulate daily profiles at $3°$ lon $\times$ $2°$ lat (DOMINO) and $0.667°$ lon $\times$ $0.5°$ lat (POMINO and DOMINO2_GC), respectively..."

For the statement "Valin et al. (2013) showed that the concentration of NO2 downwind of a city increases significantly with wind speed. on Page 3 Lines 24-25, can a range be provided to which this statement is true.
We have added specific numbers based on Fig. 4 in Valin et al. 2013:

"Valin et al. (2013) showed that the concentration of $NO_2$ downwind of a city increases significantly with wind speed, observing that $NO_2$ 100–200 km downwind from Riyadh, Saudi Arabia was approximately 130–250% greater for wind speeds between 6.4–8.3 m s$^{-1}$ than wind speeds $< 1.9$ m s$^{-1}$."

Page 3, Lines 27-29: The authors should explain why OMI is less sensitive at lower altitudes or provide a reference which explains this.
In response to the fifth comment, we added this before Eq. 1. We have included a phrase directing the reader to that explanation here:

"As discussed before Eq. (1), UV/visible satellite observations of $NO_2$ are less sensitive to $NO_2$ at low altitudes..."

In the Introduction, we are informed that this study focuses on Atlanta. The reasoning for this is explained later on in the manuscript, but needs to be mentioned here as well to make it clear why this region is the focus of the study.
We have added a sentence to page 4, around lines 21–22 that explains this reasoning:

"...to demonstrate the impact of day-to-day variations in the modeled $NO_2$ profiles on the calculated AMFs surrounding a major urban area such as Atlanta, GA, USA. Atlanta provides an example of a strong $NO_x$ area source relatively isolated from other sources, with straightforward response of the day-to-day a priori profiles to meteorological variables."

Please reword Lines 30 (P4) 2 (P5) to make the text clearer. i.e. These have been classified as the row anomaly and as of 5 July 2011 affect one-third of the pixels http://projects.knmi.nl/omi/research/product/rowanomaly-background.php), reducing coverage from global daily to global every two days.

We have reworded this to:

"It has a continuous data record since 1 Oct 2004, with global daily coverage for the first $\sim 3$ years of operation. Since 25 June 2007, anomalous radiances have been observed in several of the pixel rows. These have been classified as the "row anomaly"..."

In the title of section 2.2 please expand out what BEHR represents. Also, P5, L5, BEHR doesnt need to be expanded again as it is done so in the Introduction.

These changes have been made.

In Eqn 2,3 please specifically state what p represents. Also, how is ptp defined (e.g. dynamical, chemical tropopause)?

We have added this to lines 16–21 on p. 5 ($p$ is the vertical coordinate pressure, the tropopause is defined as a static value of 200 hPa). This paragraph has been expanded to address other reviewers' comments as well:

"...$p$ represents the vertical coordinate as pressure. $w(p)$ represents scattering weights derived from the NASA SP v2 look up table. $g(p)$ represents the mixing ratio $NO_2$ a priori profile taken from WRF-Chem, simulated at 12 km resolution in the published BEHR product. $p_0$ represents the surface pressure (clear sky AMF) or cloud pressure (cloudy AMF) of the satellite pixel, and $p_{tp}$ the tropopause pressure. The cloud pressure is that provided in the NASA SP v2 product, and is retrieved using the OMI $O_2$-$O_2$ cloud algorithm (Acarreta et al., 2004; Sneep et al., 2008; Bucsela et al., 2013). A static tropopause pressure of 200 hPa is used. $p_{surf}$ in Eq. (3) is the terrain surface pressure. The integration is carried out using the scheme described in Ziemke et al. (2001) which allows integration of mixing ratio over pressure."

Please provide a reference or explanation why "an assumed cloud albedo of 0.8" is used, on P5 , L 21.

Citations for Stammes et al. (2008) and Bucsela et al. (2013) have been added, this paragraph has been expanded for other comments as well:

"The scattering weights, $w(p)$, depend on the viewing geometry, surface albedo, and terrain pressure altitude. The BEHR algorithm uses the $0.05° \times 0.05°$ combined MODIS MCD43C3 black-sky albedo product and a surface pressure derived from the Global Land One-km Base Elevation project database (`http://www.ngdc.noaa.gov/mgg/topo/globe.html`; Hastings and Dunbar 1999) with a 7.4 km scale height as inputs to the clear sky scattering weights. Cloudy scattering weights treat the cloud pressure as the surface pressure and use an assumed cloud albedo of 0.8 (Stammes et al., 2008; Bucsela et al., 2013). The final AMF is computed as the cloud radiance fraction ($f_{rad}$) weighted average of the clear and cloudy AMFs (Eq. 4). The cloud radiance fraction is taken from the SP v2 data product (Bucsela et al., 2013)."

P5 Line 29: What does NE1 11 stand for/represent?
This stands for National Emissions Inventory 2011; the abbreviation has been expanded.

The manuscript should not reference papers in prep such as Zare, (in prep) on P6 Line 4.
We have removed this citation; this now reads:

"...the RACM2 scheme is customized to reflect recent advancements in understanding of alkyl nitrate chemistry using Browne et al. (2014) and Schwantes et al. (2015) as a basis."

P5 L5: Please provide justification for using a spin up period of just 5 days.
This is similar to Browne et al. (2014). Qualitatively, after about 24 hours, the initial conditions of $NO_2$ appear to have little effect on the modeled concentrations significantly due to the short lifetime of $NO_2$. The initial concentrations are removed chemically, and the modeled concentrations are driven by the emissions and meteorology. Other species affecting $NO_x$ chemistry, such as ozone and peroxy acetylnitrates (PANs) appear to require about 3 days before the initial concentration no longer has a large impact on the modeled concentrations. We have added a reference to Browne et al. (2014).

"The model is run from 27 May to 30 August, 2013. Similar to Browne et al. (2014), the five day period 27–31 May is treated as a spin up period, thus we use 1 June to 30 August as our study time period."

P6, L20: Please rephrase " 'pseudo-retrieval' that is much simplified compared to a full operational NO2 retrieval" and outline how it is much simpler. Also on L21, please state some examples of co-founding variables.
Both the simplification and confounding variables are addressed in the numbered points below this paragraph (P7, L15–24). We have removed the sentence mentioned in this comment, as we see how it can be confusing, and believe that by doing so, we direct the reader to these points further on in the paragraph where they are addressed in more detail. The paragraph beginning on P.6, L.20 now begins with:

"Two retrievals are used to study the effects of incorporating daily a priori profiles in the BEHR algorithm. The first is what we term a "pseudo-retrieval." To create this retrieval, an $11 \times 19$ (across $\times$ along track) subset of pixels..."

Section 2.5: Firstly, I cannot find an example of where EMG is expanded (i.e. what does it stand for)? Secondly it should be expanded in the introduction where EMG is first mentioned. It should also be written in full for the title of this section.
We have expanded it in both the section title and introduction.

P7 L20: An explanation on why the WRF winds are transformed to earth-relative from grid-relative would be much appreciated.
We have added a concise explanation to P.7, L.16–20:

"The surface wind direction and speed are calculated as the average of the first five layers ($\sim$ 500 m) of the 9 WRF 12 km grid cells closest to Atlanta at 1400 local standard time for each day. WRF wind fields are given relative to the model grid; however, the $x$ and $y$ coordinates of the grid do not correspond directly to longitude and latitude. Therefore, the wind fields must be transformed from grid-relative to earth-relative..."

Eqn 8: Please define specifically what F(x. . ...) represents. Also stating what "erfc" stands for would be useful as well. Im assuming it is "error function"?
We have expanded the paragraph between Eq. 8 and 9 to address both of these concerns, as well as better connect these two equations for the reader:

"where erfc is the error function complement, i.e. $\mathrm{erfc}(x) = 1 - \mathrm{erf}(x)$. $F(x|a, x_0, \mu_x, \sigma_x, B)$ serves as an analytical function that can be fitted to the observed line densities. We find the values of $a$, $x_0$, $\mu_x$, $\sigma_x$, and $B$ that minimize the sum of squared residuals between $F(x|a, x_0, \mu_x, \sigma_x, B)$ and the line densities, $NO_2(x)$:"

P9, L8-9: As stated above, the reasoning for choosing Atlanta needs to be outlined earlier on in the manuscript.
This was addressed at the previous comment.

In Figure 2b, it would be useful to add a scale for the wind speed or add some text to the caption stating what the min, mean and max winds in the domain are for that period.
We have added these statistics to the caption, which now reads:

"Average conditions for June 2013. (a) The red box indicates the part of the SE US being considered. (b) Surface wind directions from the WRF model; average wind speed is 5.0 m s$^{-1}$ (min 1.7 m s$^{-1}$, max 12.7 m s$^{-1}$). (c) WRF-Chem tropospheric $NO_2$ columns. (d) AMFs for the pseudo-retrieval calculated using the average monthly $NO_2$ a priori. The direction of the colorbar is reversed in (d), as small AMFs correspond to high modeled VCDs. In all panels, the star ($\bigstar$) indicates the position of Atlanta. Longitude and latitude are marked on the $x$- and $y$- axes, respectively."

P9-10, L30-1: Please expand on "All pixels show a positive change." Is this correct. Should there not be negative changes somewhere in the domain?

We were also surprised initially that all pixels showed a positive change. The remainder of that paragraph (P10, L1–5 in the original paper) explains the cause of this positive everywhere change. We have added a connective clause that makes it clearer that the rest of the paragraph addresses this point:

> "All pixels show a positive change. This occurs because 77% of the daily profiles have less $NO_2$ than the corresponding monthly average profile..."

Please expand VCDs in the section 3.2 title

We have expanded the definition of VCDs in the section 3.2 title.

P11, L9-11, the authors state that the uncertainty value of 1015 molecules per cm2 can be reduced by a factor of the SQRT(n). However, this assumes that all errors in this uncertainty value are random. Surely, some of the error will be systematic or smoothing errors? Therefore, the authors show take this into account or explain why it can be done like this.

Another reviewer also made this point. Boersma et al. (2004) notes that an explicit separation of the random and systematic components is very difficult, so the errors are usually treated as entirely random. Nevertheless, while we retain the $1 \times 10^{15}$ molec. cm$^{-2}$ criteria for the discussion because of its simplicity, we also added a second criterion that separates error due to spectral fitting, stratospheric separation, and AMF calculation. We assume that the spectral fitting and stratospheric subtraction errors are random, and half of the AMF error. The expanded paragraph is:

> "Implementing the daily profiles also changes the average VCDs, in addition to the day-to-day changes in VCDs discussed above. Figure 4b shows the changes in VCDs averaged over the period studied. The largest decrease around Atlanta is to the northeast, along the direction that the monthly average model results placed the $NO_2$ plume, but clear decreases can also be seen to the northwest and southwest. In these directions, a systematic decrease of up to 8% ($4 \times 10^{14}$ molec. cm$^{-2}$) is observed. Although this change is small, it is expected to be systematic. Statistically, a pixel's a priori profile is more likely to have less surface $NO_2$ when different wind directions are no longer averaged in, thus decreases in the VCD when using a daily a priori profile are more common.
>
> Greater relative changes are observed around the smaller cities of Birmingham (down to $-12.5\%$, $5 \times 10^{-14}$ molec. cm$^{-2}$) and Montgomery (down to $-13\%$, $4 \times 10^{-14}$ molec. cm$^{-2}$). This appears to be due primarily because the areas of emissions are smaller which makes shifts in wind direction have a greater average relative effect on the plume shape.
>
> We also compare this average change to the measurement uncertainty. The uncertainty due to random errors in the retrieval should reduce as the square root of the number of observations, but delineating random and systematic errors

in the retrieval is challenging (Boersma et al., 2004). The most optimistic approach assumes that the global average uncertainty of $1 \times 10^{15}$ molec. cm$^{-2}$ (Bucsela et al., 2013) can be treated entirely as random error, and can be reduced by $\sqrt{40}$ for the number of observations (not impacted by clouds or the row anomaly), to a lower bound of $\sim 1.6 \times 10^{14}$ molec. cm$^{-2}$. Most of the changes near the three cities exceed this lower limit. More realistically, the spectral fitting and stratospheric uncertainty may be considered largely random, but only part of the error in the AMF calculation is random, due to spatial or temporal autocorrelation in the models or ancillary products (Boersma et al., 2004). For simplicity, we assume that the spectral fitting and stratospheric subtraction errors are entirely random, while only half of the error in the AMF is random. This reduces the error from $\sqrt{(0.7 \times 10^{15})^2 + (0.2 \times 10^{15})^2 + (20\%)^2}$ to $\sqrt{(0.11 \times 10^{15})^2 + (0.03 \times 10^{15})^2 + (11.6\%)^2}$. Only the largest changes near Birmingham and Montgomery exceed this threshold. This more conservative estimate suggests that the changes in averages are primarily important for smaller or very geographically concentrated cities, where wind direction can have a large effect. Nevertheless, larger cities may exhibit important changes as well."

The two sentences on P11, L7-9 "The main decrease around Atlanta is to the northeast, along the direction that the monthly average model results placed the NO2 plume. A systematic decrease of 5-10% to the northeast of Atlanta is observed; this is the plume direction in the monthly average profiles." need to be reworded as discussing "northeast" twice is repetitive.

This has been done; we've also better indicated that decreases are seen in other directions as well (see first paragraph of revision for previous comment).

Just double checking on P11, L17, this should definitely "southeast"?

Yes, the wind blows to the southeast most frequently. The average wind direction for June 2013 is to the northeast, but this is because the wind direction usually falls within a 180° arc centered on the northeast.

P12, L12. Should be "a x0" and not "an x0".

According to the Chicago Manual of Style, "an" before the "ex" sound is acceptable, and we prefer that. (`http://www.chicagomanualofstyle.org/qanda/data/faq/topics/Usage/faq0068.html`)

On P13, L14-19, the t-test is discussed to determine if "differences in emissions and lifetimes are significantly different among the results derived from using the three different a priori profiles...." The t-test assumes that data within the sample population are independent. However, I imagine there will be lots of temporal autocorrelations in the samples. Do the authors account for this and if not, why?

We agree with the reviewer that it is likely that there would be some degree of autocorrelation in the temporal evolution of VCDs over the study period, and so in the day-by-day line densities as well. However, the emissions and lifetime are derived from fitting parameters that fit the dependence of the temporal average line densities as a function of space. Of

course, there will naturally be spatial autocorrelations as well, given the physical processes governing the evolution of the $NO_2$ plume. The EMG function used to fit these line densities should account for these physical processes (Beirle et al., 2011), and according to Chatterjee and Hadi (2012), autocorrelation often appears when the fitting model does not include dependence on a key variable. Since a Durbin-Watson test indicates that there is still unaccounted for spatial autocorrelation, we now acknowledge in the paper that the $t$-tests may underestimate the uncertainty and focus on the fact that these changes will be systematic:

> "We also use 2-sample $t$-tests at the 95% confidence level (Harris, 2010) to determine if differences in emissions and lifetimes given in Table 3 are significantly different among the results derived from using the three different a priori profile sets for a given city and wind speed bin (i.e. we compare the three values of emissions derived using different a priori profiles for Atlanta and wind speeds $\geq 3$ m s$^{-1}$). This found that, for emissions, the choice of a priori leads to statistically different emissions for all five cases. For the derived lifetimes, in all cases the monthly 108 km and daily 12 km a priori are statistically indistinguishable, but the monthly 12 km a priori is statistically different. We note that a Durbin-Watson test indicates some spatial autocorrelation remains, and so the uncertainty may be underestimated and the $t$-tests may be incorrectly identifying the differences as significant in this case (Chatterjee and Hadi, 2012). Even if this is true, with a longer averaging period such as those in Beirle et al. (2011), Valin et al. (2013), and Lu et al. (2015), we would expect the random uncertainties to reduce while the systematic difference from the choice of a priori profile remains. Therefore, the choice of a priori profiles does have an important effect on derived emissions and lifetimes."

On P14, Lines 3-15, comparisons to the NEI 11 emissions are discussed. From the text, the NE1 11 emissions are overestimated by 50%. If the NEI 11 emissions took this into account, which results (e.g. monthly or daily apriori ) which have best agreement with them. Also, on L7, the authors state that the "daily 12 km a priori are within 5- 24%". Are they lower or higher than the NEI 11 emissions or both? The authors have stated that the "coarse monthly a priori" are lower by 43-62%.

We have reorganized this paragraph to accommodate this comment; the sentence indicating which emissions agree with the current and 50% reduced NEI emissions has been brought closer to the statement about the uncertainty in the NEI inventory. That emissions derived from daily 12 km profiles are both greater and less than the NEI inventory has been explicitly stated:

> "We also compare the derived emissions rates to the emissions in a 12 km WRF-Chem model driven by the NEI 11 emission inventory with $NO_x$ emissions scaled to 88.9% of the 2011 values to account for the decrease between 2011 and 2013 (EPA, 2016). WRF-Chem emissions are calculated as the sum of all grid cells within a 50 km radius of the city. 50 km was chosen as the line densities were integrated for $\sim 50$ km to either side perpendicular to the wind direction. The coarse monthly a priori are 43–61% lower than the NEI-driven emissions, while

emissions derived using daily 12 km a priori are within 5–24% (both greater and less than the NEI emissions). Recent work (e.g. Travis et al. 2016 and references within) suggests that the NEI inventory is overestimated by $\sim 50\%$ using both satellite and in situ observations. Emissions derived using daily 12 km show the best agreement to the current NEI inventory, and emissions derived using monthly 108 km a priori profile agree with the NEI inventory reduced by 50%. Therefore, we cannot say which a priori profiles provide the best measurement of emissions by comparing to NEI...."

**Supplement**

P1, L16: Why would we expect the mean and median differences to be 0?

Since the upper troposphere is generally removed from $NO_x$ sources, we expect $NO_2$ concentrations there to be fairly stable. Deviations from the average should mostly be due to larger scale motions of air masses, also separated from direct influence from $NO_x$ sources, so the deviations should be essentially random and normal. Only at the surface with direct influence of $NO_x$ sources do we expect a non-normal distribution of $NO_2$ concentrations, and so a non-zero mean difference in the average of AMF calculated from the monthly average profile vs. the daily profiles.

P1, L20-22: "This, combined with the greater scattering weights at these altitudes, explains why the effect on the AMF is as large as it is, although day-to-day changes in the boundary layer still dominate the effect using daily profiles has on the AMFs,". The "," at the end of the sentence should be a full stop. Secondly, can the impact of the scattering weights at this altitude be quantified, say in comparison to the impact of the apriori on the AMFs?

Thank you again for careful proofreading. As far as quantifying the effect of the scattering weights, the response of the upper tropospheric scattering weights to perturbations in the inputs to the radiative transfer model that computes them could be calculated, but is not particularly relevant to this part of the work. More important here is that scattering weights in the UT are 2–8x higher than near the surface, so small perturbations in the $NO_2$ profile have larger effects on the AMF than the same perturbations would have near the surface. We have included this factor of 2–8x:

> "Day-to-day changes in the free tropospheric a priori profile are smaller in magnitude than those in the boundary layer, but usually occur over a much greater vertical extent. Further, the scattering weights are greater at these altitudes ($\sim$2–8 times those near the surface), amplifying the effect of small changes in the profile shape at these altitudes. This explains why the effect on the AMF is as large as it is, although day-to-day changes in the boundary layer still dominate the effect using daily profiles has on the AMFs."

P1, L30: Is this period of 91 days long enough for the averaging to have no net impact?

Yes, as evidenced by the fact that the average difference between the hybrid and full profiles is 0 (Fig. S3).

P2, L 9: Should be "Atlanta, nevertheless" or "Atlanta. Nevertheless".
Corrected, thank you.

P4, L4-5: Should be these "]" brackets and not ")"?
No, in a range, infinity should be accompanied by a round parenthesis since no finite range can truly include infinity.

P5, L20: "as the errors contributing to it should be random in nature". Why is this the case? No systematic errors?
This is similar to a previous comment on the uncertainty in VCDs; while there is definitely some systematic component, the errors are usually treated as random in practice (Boersma et al., 2004). Removing the factor of $\sqrt{n}$ does not alter our conclusions however, because we calculated $n$ conservatively, so we will use the flat 25% from Lu et al. (2015). Table 4 and this section have both been updated accordingly (P.5 L.19–24 of supplement removed).

Updated Table 4:

[revised manuscript text omitted]

---

## Author Comment (AC2) · 21 Oct 2016

**Effects of daily meteorology on the interpretation of space-based remote sensing of NO$_2$**
**Response to Anonymous Referee #2**

Joshua L. Laughner, Azimeh Zare, and Ronald C. Cohen

October 20, 2016

We thank the reviewer for their positive response and very careful reading of the manuscript. The reviewer's comments will be shown in red, our response in blue, and changes made to the paper are shown in black block quotes. Unless otherwise indicated, page and line numbers correspond to the original paper. Figures, tables, or equations referenced as "R$n$" are numbered within this response; if these are used in the changes to the paper, they will be replaced with the proper number in the final paper. Figures, tables, and equations numbered normally refer to the numbers in the original discussion paper.

While I see the nice qualitative discussion of the effects as a strong point of this manuscript, the quantitative results are much less convincing in my opinion. The reason for my skepticism is the large change in emissions and lifetimes the authors find when changing their a priori spatial or temporal resolution in combination with the large uncertainties given in the tables. Most of the results are in agreement within uncertainties when changing from monthly to daily a priori in spite of the large changes seen. To me this indicates that the time period used for averaging is too short to really separate the effect of a priori changes from noise in the data, and as the authors acknowledge, all previous studies used much longer averaging times. Please comment on the magnitude of uncertainties and the significance of differences seen.

We do understand the reviewer's skepticism, given the magnitude of the uncertainties. In the original paper, we had already included statistical tests of the significance of these differences using $t$-tests (P14, L18–23) which showed that, in most cases, the differences due to the choice of a priori profiles are statistically significant at the 95% confidence level, except for lifetimes derived using the monthly 108 km and daily 12 km profiles. Reviewer #1 raised the possibility of autocorrelations in the data, which would indicate that the uncertainties were underestimated. While we do not believe that temporal autocorrelations are significant in this method, some spatial autocorrelation seems to be unaccounted for by the EMG fit. In response to Reviewer #1, we have acknowledged this possibility in the revised paper, but note that the differences are systematic, and therefore are not expected to be reduced by additional averaging, while the uncertainties will be. Added at P.13, L.19:

> "We note that a Durbin-Watson test indicates some spatial autocorrelation remains, and so the uncertainty may be underestimated and the $t$-tests may be

incorrectly identifying the differences as significant in this case (Chatterjee and Hadi, 2012). Even if this is true, with a longer averaging period such as those in Beirle et al. (2011), Valin et al. (2013), and Lu et al. (2015), we would expect the random uncertainties to reduce while the systematic difference from the choice of a priori profile remains. Therefore, the choice of a priori profiles does have an important effect on derived emissions and lifetimes."

In the discussion of the results both in the text and in the abstract, I'm confused by statements such as "Comparing an optimized retrieval to a more standard one, we find that NOx emissions estimated from space-based remote sensing can increase by 100% when daily variations in plume location and shape are accounted for in the retrieval." If I'm not misinterpreting Table 4, the change in emissions when moving from monthly to daily a priori is closer to 45% and actually is a decrease, not an increase of estimated emissions. This also makes more sense considering the qualitative discussion given in the first part of the paper. The factor of 2 increase is relative to a low spatial resolution a priori which is also interesting but not the focus of the study and also not what is suggested by the formulations in the text. I think these statements need to be rephrased.

Thank you for clearly explaining what you found confusing about this statement; we agree now that its present form is unclear. We have reworded it in both the abstract and conclusions to clearly indicate that there are different effects of spatial and temporal resolution on the emissions. While the effect of temporal resolution is indeed the main focus of this paper, it is important to separate the effects of spatial and temporal resolution since no application of the EMG fitting method has yet used a retrieval with high spatial resolution profiles (Beirle et al., 2011; Valin et al., 2013; Lu et al., 2015; Liu et al., 2016).

In the abstract, P.2 L.8–9 changed to:

> "Additionally, we show that $NO_x$ emissions estimated from space-based remote sensing using daily, high spatial resolution a priori profiles are $\sim 100\%$ greater compared to a retrieval using spatially coarse a priori profiles, and 20–45% less compared to a retrieval using monthly averaged high spatial resolution profiles."

And in the conclusion, P.15 L.3–5 changed to:

> "When the methods of Lu et al. (2015) are applied to these prototype retrievals, significant changes in derived $NO_x$ emissions are found, increasing by as much as 100% for Atlanta compared to emissions derived from a retrieval using coarse a priori profiles."

The discussion of standard mathematical methods for fitting a function to the decay curve is a bit out of place in such an article, in particular as the method used for emission estimation is not the topic of the paper. I would suggest to shorten this part and to remove the discussion of Matlab internals which are of little interest to the reader.

Both other reviewers also requested that references to Matlab functions be made more general, which we have done. The technical considerations have also been moved to the supplement (P.8, L.9–29).

The approach taken to averaging the model data in time (Equation 5) appears overly complex and not transparent. The obvious way to treat this problem is simple interpolation in time to the OMI overpass.

Effectively this approach is an interpolation, as an interpolation can be recast as a weighed average. If we are interpolating to a point $(x_q, y_q)$ between $(x_1, y_1)$ and $(x_2, y_2)$:

$$
\begin{aligned}
y_q &= y_1 + m(x_q - x_1) \\
&= y_1 + \left( \frac{y_2 - y_1}{x_2 - x_1} \right)(x_q - x_1) \\
&= \frac{x_q - x_1}{x_2 - x_1} y_2 + \left( 1 - \frac{x_q - x_1}{x_2 - x_1} \right) y_1 \\
&\equiv w y_2 + (1 - w) y_1 \text{ where } w = \frac{x_q - x_1}{x_2 - x_1}
\end{aligned}
$$

This approach assumes that the average OMI overpass time varies linearly with longitude. We have added a paragraph after Eq. (5) and a section to the supplement explaining this reasoning; essentially, as shown in Fig. R1, there is the possibility that an area (such as Atlanta) may be covered by the west edge of an earlier swath or the east edge of a later swath, so these weights are an *ad hoc* attempt to choose profiles that approximate the average overpass time.

After Eq. (5):

> "The weighting scheme in Eq. (5) was chosen over simply using the model output for 1400 local standard time for each latitude to create smooth transitions between adjoining time zones. This attempts to account for the day-to-day variability in OMI overpass tracks as well as the fact that pixels on the edge of a swath can be observed in two consecutive overpasses at different local times. More detail is given in the supplement."

And in the supplement:

> "When computing the monthly average profiles, it is necessary to use profiles that represent OMI's overpass time, typically quoted as 13:30 to 13:45 local standard time (e.g. McLinden et al. 2014; Levelt et al. 2006). To average the profiles output from WRF-Chem, weights were calculated that fulfilled two requirements:
>
> 1. The weights should be 1 at OMI overpass time and 0 when more than 1 hour away from overpass time.
> 2. The transition between profiles from different hours should be smooth.
>
> For #1, we assume that the average overpass time is 1330 local standard time. We compute local standard time as:

$$
t_{\text{apriori, local}} = \frac{l}{15} + t_{\text{apriori, utc}} \tag{R1}
$$

[Figure]

Figure R1: Swaths covering the east coast of the US for 2 June 2013 (a,b) and 3 June 2013 (c,d). The times given are the start and end times of the daytime half of the orbit in UTC. As shown, on different days, the time of the OMI swath that covers Atlanta can vary by up to an hour.

where $t_{\text{apriori, local}}$ is the local standard time in hours past midnight, $t_{\text{apriori, utc}}$ the UTC time in hours past midnight, and $l$ the longitude (west is negative). To meet the second requirement, this is a continuous function, rather than a step function (where each 15° longitudinal segment/time zone has a single local time). Areas further west in a time zone are more likely to be observed on the east edge of a later OMI swath, and vice versa for areas further east. This weighting includes some influence from later profiles to account for this.

The weights from Eq. (5) are derived from:

$$w = 1 - |t_{\text{overpass}} - t_{\text{apriori, local}}| = 1 - \left| 13.5 - \frac{l}{15} - h \right| \qquad (\text{R2})$$

where $t_{\text{overpass}}$ is the assumed overpass time for OMI and $h \equiv t_{\text{apriori, utc}}$. If $w < 0$, $w$ is set to 0. This gives us the desired form where the weights smoothly vary in time.
"

In the course of revisiting this weighting method, we redid the key runs using an assumed OMI overpass time of 1330 LST (which is more consistent with previous papers, e.g. McLinden et al. 2014, compared to 1400 LST used originally). The difference is very small and does not affect the conclusions of the paper, but we have updated the appropriate figures and parts of the text.

On a more general note I think that the paper would benefit from a short discussion of the impact model errors could have on the results. It is obvious that in theory, using daily a priori data is better than using monthly averages as the process of NO2 retrieval is not linear. However, in real data this might not necessarily be true. At the high spatial resolution of the model used here, even a small uncertainty in wind direction, emission height or emission source location can move the NO2 plume into different model grid cells, potentially leading to poor matching of plume position in measurement and model and thus wrong air mass factors and NO2 columns. I find this an interesting topic in particular in view of future instruments having improved spatial resolution.

Agreed, model uncertainty is an important (and deep) topic highly relevant to this paper, so we have added a section to the discussion with our thoughts on this topic. There has been some interesting work comparing WRF-Chem simulations to measurements both directly (Tie et al., 2007; Zhang et al., 2009) and in terms of spatial variability (Follette-Cook et al., 2015) which suggest WRF does capture daily wind fields and $NO_2$ distribution reasonably accurately. We would also expect the overall uncertainty to change source, but not necessarily magnitude between monthly and daily profiles, as monthly profiles effectively have the true day-to-day variability as a source of uncertainty that daily profiles do not; but in exchange, the random error in the monthly profiles is reduced through averaging. We have added a new section at the beginning of the discussion that covers this point:

> "WRF-Chem has generally been found to reproduce wind fields, especially above $2 \text{ m s}^{-1}$ (Tie et al., 2007; Zhang et al., 2009), and spatial variability of trace gases (Follette-Cook et al., 2015) well. Nevertheless, a natural concern when modeling daily $NO_2$ profiles for satellite retrievals is the accuracy of the plume location. We, however, note that the transition from monthly average to daily profiles does not necessarily result in increased model uncertainty, but rather a change in the type of uncertainty.
>
> When using monthly average profiles, the uncertainty in the modeled $NO_2$ concentrations compared to the true mean will be reduced (assuming at least some component of the error is random in nature), but the true day-to-day variability not captured by the monthly average effectively becomes a new error term. In contrast, when using daily profiles, the random model error is not reduced, but the day-to-day variability is also not averaged out. Ideally, the error in a set of daily profiles will manifest as deviation from the true set of profiles for that day, rather than the monthly profiles' smaller deviation from a mean set of profiles that itself may not represent any single day.
>
> An important step in managing the uncertainty in the daily profiles is to constrain the modeled meteorology with observations or reanalysis datasets. By default, meteorology in WRF is constrained via initial and boundary conditions only. With larger domains and longer runs, further constraints using four-dimensional data assimilation (FDDA, Liu et al. 2006) and/or objective analysis (Follette-Cook et al., 2015; Wang et al., 2014; Yegorova et al., 2011), possibly combined with periodic model reinitialization (Otte, 2008) are strongly recommended."

Page 7, line 5: Cloudy AMF is smaller, not lager [larger] for boundary layer NO2 profiles
Thank you for bringing this up; we double checked this statement by comparing clear sky and cloudy AMFs from both the NASA standard product and our own, and found that in both products, the regression slope of cloudy vs. clear sky AMFs is $> 1$, but that there is sufficient scatter that it is hard to say definitively that one is larger or smaller. Thus, we've reworded this point to simply identify that clear and cloudy AMFs can be significantly *different*, but not that one is necessarily greater than the other. P.7, L.5 changed to:

> "Day-to-day variations in cloud fraction also lead to large changes in AMF because the presence of clouds changes both the scattering weights (due to high assumed reflectivity of clouds and smaller effective surface pressure compared to ground) while also obscuring the $NO_2$ profile below the cloud."

Page [15] line 6: I do not understand the sentence Further work is needed to understand the impact of this change on top down constraints of NOx emissions, given the recent work showing that bottom up estimates are high by 50%. I think the relevance is obvious if you believe your own results all previous estimates based on this technique and using monthly high resolution a priori profiles give too high emission estimates. Whether or not the emission inventory is off is another topic (which could of course also impact on your AMF values and thus emission estimates) and should not be mixed here.
Indeed, any previous work using monthly high resolution profiles would be biased high, which would fit in nicely with Travis et al. (2016) indicating that the bottom up inventory is too high. However, the previous application of the EMG method to US emissions used a retrieval with coarse a priori profiles (Lu et al. 2015); our results indicate that emissions found using a retrieval with spatially coarse set of a priori profiles will be lower than when using daily, fine spatial resolution profiles. Hence the need for more work–Travis et al. (2016) found that the bottom up inventory should be reduced by $\sim 50\%$ using GEOS-Chem as a transfer standard between aircraft and satellite measurements, while we find that, with better a priori, the bottom up inventory is about 25% high.
We have edited this sentence to make this clearer, it now reads:

> "Emissions derived using the fine daily a priori are within 25% of the bottom up number from the NEI inventory, a smaller reduction than that suggested by Travis et al. (2016). Future work will aim to resolve this difference."

Finally, each of the following technical issues has been addressed

Equation 2: $w_s(p)$ is later written as $w(p)$

Equation 4: Please add how cloud radiance fraction was computed
Added just before Eq.(4):

> "The cloud radiance fraction is taken from the SP v2 data product (Bucsela et al., 2013)."

Page 6: Add reference for MOZART model
Added Emmons et al. (2010).

Page 14, line 17: something missing here?
The "and" after "$a$" was unnecessary.

Conclusions, line 28: Please add again that you count days even if only a single pixel shows a change larger than the noise (which I personally find a strange way of counting)

[revised manuscript text omitted]

---

## Author Comment (AC3) · 21 Oct 2016

**Effects of daily meteorology on the interpretation of space-based remote sensing of NO$_2$**
**Response to Anonymous Referee #3**

Joshua L. Laughner, Azimeh Zare, and Ronald C. Cohen

October 21, 2016

We thank the reviewer for their helpful comments, especially the points of clarification in the introduction. The reviewer's comments will be shown in red, our response in blue, and changes made to the paper are shown in black block quotes. Unless otherwise indicated, page and line numbers correspond to the original paper. Figures, tables, or equations referenced as "R$n$" are numbered within this response; if these are used in the changes to the paper, they will be replaced with the proper number in the final paper. Figures, tables, and equations numbered normally refer to the numbers in the original discussion paper.

One limitation is the focus on only one area (Atlanta, GA) and one season (Summer). As pointed out by another reviewer, focusing on such a short period leads to a limited dataset, from which it is difficult to obtain estimates of emissions and lifetimes that are significantly different. Since the differences in the estimated emissions are at least a factor of 2, it should be feasible to achieve statistically significant differences by analysing a longer period. This would strengthen the paper considerably.

We are glad to see that the reviewer is interested in the application of daily profiles when constraining emissions and lifetime. It is definitely expected that extending the analysis to a longer time period would allow a more quantitative analysis of emissions and lifetime, but the goal of this paper was primarily to show that there is an important difference in the AMFs and therefore VCDs retrieved using daily vs. monthly profiles, and that these differences do impact the emissions and lifetime inferred from the product. While the uncertainties in the emissions and lifetimes are large, we conclude that the differences due to the a priori are systematic and describe our reasoning why. We hope to extend the time period and geographic area studied to get quantitative estimates of emissions and lifetimes across a number of US cities, and that others will keep these results in mind when considering top-down emissions and lifetime constraints.

The systematic nature of these changes is discussed on P.13 L.20–P.14–L.2. In response to another reviewer's comment, this paragraph has been extended starting from P.13 L.31:

> "In the EMG fit, this manifests as a too short lifetime. As the emissions are inversely proportional to lifetime (Eq. 11), emissions derived using the monthly 12 km a priori profiles will be too great. Therefore, when using a retrieval with a priori profile at fine spatial resolution, daily temporal resolution of the a priori

profiles is necessary to prevent underestimating the lifetime. Further, the spatial resolution of the a priori profiles has a large impact on the magnitude of the derived emissions. To reduce the systematic biases in emissions and lifetime from the choice of a priori profile, it is necessary to simulate these profiles at fine spatial and daily temporal resolution."

We have also added the following sentence on P.4, L.21 to make clear that our goal is not yet to quantitatively constrain emissions and lifetime, but to demonstrate that the systematic error due to the use of monthly average profiles is important and should be addressed:

"...meteorological variables. Our point is not to derive exact answers for the size and frequency of the effects of daily profiles, but rather to illustrate that these effects are large enough that their role should be assessed in any future analysis that does attempt to interpret space-based remote sensing of $NO_x$. We show that the variability..."

Then there is a serious error in the theoretical framework for BEHR AMFs. According to Eq. (2) and the text in section 2.2, the cloudy AMF is calculated only between the cloud pressure level and the tropopause. In principle this can be done, but then the retrieval needs to account for a so-called ghost column [Burrows et al., 1999]. A ghost column correction however, is not being applied here. The better alternative is to calculate both the clear-sky and cloudy-sky AMFs by integrating the NO2 profile from the surface pressure to the tropopause. This formulation ensures that the AMF value returns a tropospheric column that is representative for all NO2 in the troposphere, and not just for the above-cloud fraction (in the limiting case of cloud fraction 1.0). The authors should revisit their integration limits for Eq. (2), probably also for their current BEHR-product.

The reviewer is correct that this formulation of the AMF yields the visible $NO_2$ column, however this is the calculation given in the theoretical basis document for the OMI retrievals (Boersma et al., 2002). P. 20 of the TBD indicates after Eq. 2-15 that "$z$ is the altitude of the lower boundary (ground or **cloud top**)," where $z$ is the lower integration bound in the numerator of the AMF calculation. The publicly available BEHR retrieval includes a multiplicative factor which allows users to add the estimated ghost fraction in, if desired. This allows the user to choose how they want to use the product, e.g. (1) as the observed, visible column, (2) for cloud slicing approaches (e.g. Belmonte Rivas et al. 2015; Choi et al. 2014 for $NO_2$, or Ziemke et al. 2009, 2001 for $O_3$), or (3) with the ghost column included to get a full column.

We acknowledge the various ways of handling below-cloud $NO_2$ on P.2 L.25:

"Finally, the tropospheric slant column density (SCD) must be converted to a vertical column density (VCD) by use of an air mass factor (AMF) and Eq. (1). Depending on the specific algorithm, $NO_2$ obscured by clouds may be ignored (producing a visible-only tropospheric $NO_2$ column, e.g. Boersma et al. 2002), corrected by use of an assumed ghost column (e.g. Burrows et al. 1999; Koelemeijer and Stammes 1999), or corrected via the AMF (e.g. Martin et al. 2002). In all cases, the AMF must account for..."

We also added information about the available ghost column factor in the BEHR product after Eq. (4):

> "Calculating clear and cloudy AMFs and using the weighted average to compute the final AMF is consistent with the OMI algorithm theoretical basis document (Boersma et al., 2002) and yields only the visible $NO_2$ column as the final product; the visible column is the value provided in the BEHRColumnAmountNO2Trop field. A scaling factor is provided in the BEHR product for users who wish to include the ghost column. This factor, $G$, is computed as:

$$G = \frac{V_{\text{surf}}}{(1 - f_{\text{geo}})V_{\text{surf}} + f_{\text{geo}}V_{\text{cld}}} = \frac{\int_{p_{\text{surf}}}^{p_{\text{tp}}} g(p)\,dp}{(1 - f_{\text{geo}})\int_{p_{\text{surf}}}^{p_{\text{tp}}} g(p)\,dp + f_{\text{geo}}\int_{p_{\text{cld}}}^{p_{\text{tp}}} g(p)\,dp} \quad \text{(R1)}$$

> where $V_{\text{surf}}$ and $V_{\text{cld}}$ are the modeled vertical column densities above the ground surface and cloud, respectively, and which are obtained by integrating the a priori profile above the surface or cloud pressure. $f_{\text{geo}}$ is the geometric cloud fraction included in the NASA standard product, which is the OMI $O_2$-$O_2$ cloud product (Acarreta et al., 2004). This factor is stored in the BEHRGhostFraction field of the BEHR product. Multiplying the VCDs stored in BEHRColumnAmountNO2Trop by these values will provide the estimated total (visible + ghost) column.
>
> The results obtained in this work use the visible columns only. The ghost column is not added in for any of the following results."

We have clarified one point in Eq. (2) and (3). In (3), the lower bound of integration is always the surface, but in Eq. (2), the lower bound is the cloud or surface pressure, for cloudy and clear sky AMFs respectively:

> "$p_0$ represents the surface pressure (clear sky AMF) or cloud pressure (cloudy AMF) of the satellite pixel, and $p_{\text{tp}}$ the tropopause pressure...$p_{\text{surf}}$ in Eq. (R3) is the terrain surface pressure.

$$\text{AMF} = \int_{p_0}^{p_{\text{tp}}} w(p)S(p)\,dp \quad \text{(R2)}$$

> where

$$S(p) = \frac{1}{\int_{p_{\text{surf}}}^{p_{\text{tp}}} g(p)\,dp} g(p) \quad \text{(R3)}$$

> "

This error is also the reason for the incorrect statement on page 7, under point 3: ... a cloudy pixel will have a much greater AMF than a clear one. The statement would be correct only if the word AMF would be replaced by above-cloud AMF. And, in line with the above

criticism, an above-cloud AMF can only return an above-cloud NO2 columns, which cannot be the purpose of a state-of-science retrieval.

As discussed above, the choice to retrieve the visible column only matches the OMI theoretical basis document (Boersma et al., 2002), and both visible only and full column retrievals have uses. Reviewer #2 also pointed out that AMFs for cloudy pixels are not always larger than clear-sky AMFs, which we confirmed by comparing cloudy and clear AMFs from our product and the NASA standard product. Consequently, this sentence now indicates that changing cloud fraction can lead to a very different AMF, and discusses the reasons for that.

> "Setting cloud fractions to 0 ensures that the AMF for every pixel is calculated with the full a priori profile, rather than just the above cloud part. Day-to-day variations in cloud fraction also lead to large changes in AMF because the presence of clouds changes both the scattering weights (due to high assumed reflectivity of clouds and smaller effective surface pressure compared to ground) while also obscuring the $NO_2$ profile below the cloud."

There are difficulties with the interpretation of uncertainties in the tropospheric column retrieval. On page 10, the choice for 1.0x1015 molec. cm-2 as a typical number for the uncertainty in the tropospheric NO2 column is rather arbitrary. Detailed error analyses (e.g. Boersma et al. [2004]) have pointed out that the uncertainty in the tropospheric column is highly variable because of AMF (a multiplicative factor indeed) uncertainties, which depend strongly on variable clouds, surface albedo, and NO2 profile shapes. Moreover, the estimate used by the authors is rather optimistic. More realistic uncer- tainties are on the order of 1.0x1015 molec. cm-2 +25% (of the individual column value).

The value of $1.0 \times 10^{15}$ molec. cm$^{-2}$ was given in Bucsela et al. (2013) as the global average mean clear sky uncertainty based on their error analysis. Since the uncertainty in the AMF is a multiplicative factor, this error may be low for urban signals. We had chosen initially to use this uncertainty, however, because we are introducing new choices for a priori profiles, which may alter the magnitude of each specific element of the uncertainty, and so having a simple, previously determined number to compare to would simplify this for the reader.

To address the reviewer's concern, we have added a new column to what was Table 2 that uses an uncertainty that is the quadrature sum of uncertainty from the slant column fitting, stratospheric subtraction, and AMF, which is $\sqrt{(0.7 \times 10^{15})^2 + (0.2 \times 10^{15})^2 + (20\%)^2}$. The uncertainty in the AMF and stratospheric separation come from Bucsela et al. (2013), which describes the error analysis for the current version of the NASA SP retrieval, and the slant column fitting from Boersma et al. (2011), cited by Bucsela et al. (2013). We add them in quadrature following Boersma et al. (2004) and Bucsela et al. (2013). This leads to the following changes in text:

P.10, L.23–29 changed to:

> "Table 2 describes how frequently significant changes in the retrieved VCD occur for pixels within 50 km of Atlanta, Birmingham, and Montgomery. Changes are considered significant by two different criteria. First, we consider the global mean clear-sky uncertainty from Bucsela et al. (2013). As we are modifying the a priori profiles, and thus potentially the uncertainty associated with the choice

of profiles, this gives us a fixed value to compare against. Second, we use the quadrature sum of uncertainties from spectral fitting ($0.7 \times 10^{15}$ molec. cm$^{-2}$, Boersma et al. 2007, 2011), stratospheric separation ($0.2 \times 10^{15}$ molec. cm$^{-2}$, Bucsela et al. 2013), and AMF calculation (20%, Bucsela et al. 2013), assuming that these are independent and so can be added in quadrature (Boersma et al., 2004). We consider the fraction of days with at least one pixel exhibiting a significant change in VCD (rather than the fraction of pixels) because the main NO$_2$ plume may only fall within a small number of pixels. Up to 54% of days exhibit changes in the VCDs greater than $1 \times 10^{15}$ molec. cm$^{-2}$, and up to 43% exhibit changes greater than the quadrature sum of uncertainties. This indicates that when considering individual daily measurements, a considerable fraction of days with any valid pixels would have biases in the retrieved VCDs above the uncertainty due to the temporal resolution of the a priori NO$_2$ profiles. "

P.11, L.3–5 changed to:

"When considering changes to be significant if they exceed $1 \times 10^{15}$ molec. cm$^{-2}$, Montgomery has the least frequent significant changes because it has the smallest VCDs, so a change to the AMF needs to be rather large to produce a significant change in the VCD by this metric, since the AMF is a multiplicative factor. When considering the quadrature sum of errors as the significance criterion, Montgomery and Atlanta both demonstrate significant changes $\sim 20\%$ of the time."

And the update to Table 2:

|  | Percent of days with $\Delta$VCD $> 1 \times 10^{15}$ molec. cm$^{-2}$ | Percent of days with $\Delta$VCD $> [\sum_i \sigma_i]^{1/2}$ | Min. change (molec. cm$^{-2}$) | Max. change (molec. cm$^{-2}$) |
|---|---|---|---|---|
| Atlanta | 39% | 23% | $-2.4 \times 10^{15}$ | $+2.5 \times 10^{15}$ |
| Birmingham | 54% | 43% | $-3.8 \times 10^{15}$ | $+3.9 \times 10^{15}$ |
| Montgomery | 27% | 20% | $-2.2 \times 10^{15}$ | $+1.9 \times 10^{15}$ |

Table 1: Statistics on the frequency and magnitude of changes in the retrieved VCDs using a daily vs. monthly average profile for pixels with centers within 50 km of Atlanta, GA, USA (84.39° W, 33.775° N), Birmingham, AL, USA (86.80° W, 33.52° N) and Montgomery, AL, USA (86.30° W, 32.37° N). The "percent of days" values are calculated as the number of days with at least one pixel in that subset with a change greater than the given uncertainty divided by the number of days with at least one pixel unobscured by clouds or the row anomaly. The uncertainty represented by $[\sum_i \sigma_i]^{1/2}$ is the quadrature sum of uncertainties from spectral fitting ($0.7 \times 10^{15}$ molec. cm$^{-2}$, Boersma et al. 2007, 2011), stratospheric separation ($0.2 \times 10^{15}$ molec. cm$^{-2}$, Bucsela et al. 2013), and AMF calculation (20%, Bucsela et al. 2013).

The specific corrections below have all been addressed. We thank the reviewer for their careful reading, and especially for the additional citations for custom retrievals, as we hope

that others will find this list a useful reference for those interested in custom satellite retrievals.

Abstract, line 6: this paper does not address variations of NO2 in power plant plumes, so this should be removed. The paper is about NO2 variations in urban plumes.
Removed.

Introduction, lines 18 and 30: a reference to the work by Vinken et al., ACP, 2014 on ship emissions estimates using improved-resolution a priori profiles would be appropriate here.
Added Vinken et al. (2014), thank you for the suggestion.

P3, lines 7-8: earlier studies by McLinden et al. [2012] showed that the oil sand signals was detectable also for retrievals using coarse-gridded a priori profiles. The statement should be nuanced in that the spatial signatures are more realistically resolved with higher resolution a priori profiles.
This sentence has been changed to:

> "McLinden et al. (2014) showed that using 15 km resolution profiles increased the $NO_2$ signal of the Canadian oil sands by $\sim 100\%$ compared to the DOMINO and NASA SP products, which they state corrects a low bias in the retrieved column amounts."

P3, L13: also include here a reference to the DOMINO retrievals using 0.5 x 0.67 profiles from GEOS-Chem over Europe from Vinken et al. [2014]. These retrievals also use diurnal profiles.
Added:

> "Conversely, the DOMINOv2 (Boersma et al., 2011), POMINO (Lin et al., 2015), and DOMINO2_GC (Vinken et al., 2014) retrievals simulate daily profiles at 3° lon × 2° lat (DOMINO) and 0.667° lon × 0.5° lat (POMINO and DOMINO2_GC), respectively, which is insufficient to capture the full spatial variability of $NO_2$ plumes, but does capture large scale variations in meteorology."

P3, L17: the NASA retrieval is usually indicated as the Standard Product v2 or SP v2. OMNO2 (actually OMNO2A v1) refers to the software for DOAS spectral fitting.
Corrected, for all instances as well as this one.

P4, L30: 13 x 24 km2
Corrected.

P4, L32: 'theoretical daily global coverage is a strange term. Coverage was nearly global every day until the row anomaly, and after that, coverage is more or less global every 2 days.
Rephrased to:

"It has a continuous data record since 1 Oct 2004, with global daily coverage for the first $\sim$ 3 years of operation. Since 25 June 2007..."

P5, L6: please provide the name and appropriate version of the NO2 SCD product used. I think it is OMNO2A v1. This product has recently been evaluated in Marchenko et al. [2015] and van Geffen et al. [2015]. It would be appropriate to cite those papers here.
Thank you for pointing us to the proper identification. This section now reads:

"Briefly, the BEHR retrieval is based on the NASA Standard Product v2 (SP v2) retrieval (Bucsela et al., 2013). The total slant column densities (SCDs) are from OMNO2A v1.2.3 (Boersma et al., 2002; Bucsela et al., 2006, 2013), and have been recently evaluated by van Geffen et al. (2015) and Marchenko et al. (2015). The stratospheric subtraction and destriping used is that of the NASA SP v2 retrieval. The tropospheric AMF is then recalculated..."

P5, L17: apart from albedo, please also provide details on the cloud information (effective cloud fraction, pressure) used in the BEHR-approach.
We have added information about the cloud pressure and cloud radiance fraction used in the retrieval. For cloud pressure, the sentence "The cloud pressure is that provided in the NASA SP v2 product, and is retrieved using the OMI $O_2$-$O_2$ cloud algorithm (Acarreta et al., 2004; Sneep et al., 2008; Bucsela et al., 2013)," was added before Eq. 2. For cloud radiance fraction, the sentence "The cloud radiance fraction is taken from the SP v2 data product (Bucsela et al., 2013)," was added before Eq. 4.

P5, L27: which version of the WRF-Chem model is used?
Version 3.5.1. The first sentence of sect. 2.3 now reads:

"Modeled $NO_2$ a priori profiles are simulated using the WRF-Chem model v3.5.1 (Grell et al., 2005)."

P6, section 2.3: please provide some more details on the WRF-Chem model such as what is the NOx emission total over the US in the period of interest, and on the meteorological and chemical boundary conditions used. It is unclear how realistic WRF-Chem simulations of NO2 profiles are. Have these been validated against surface and aircraft measurements of NO2?
The chemical mechanism used has been validated against surface data from the SOAS campaign. However, the purpose of this paper is to illustrate that using daily profiles at high spatial resolution can and does alter the retrieved $NO_2$ columns. We feel that it is important to demonstrate that the temporal resolution of the profiles is important first, and because the monthly and daily 12 km *a priori* profiles are taken from the same model run (and the 108 km profiles from a run using the same inputs at different resolution), any error in the model is in all three sets of *a priori* profiles. In this way, our goal is to show that daily, high spatial resolution profiles are important in general.
We have added details of the meteorology, boundary conditions, and emissions to the paragraph on P.5 L.26–P.6 L.4. It now reads:

"Modeled $NO_2$ a priori profiles are simulated using the WRF-Chem model v3.5.1 (Grell et al., 2005). The domain is 81 (east-west) by 73 (north-south) grid cells centered on 84.35° W, 34.15° N on a Lambert Conformal map projection (approximate edges of the domain are 89.5° W–79.2° W and 30.3° N–38° N). Meteorological initial and boundary conditions are driven by the North American Regional Reanalysis (NARR) dataset. Anthropogenic emissions are taken from the National Emissions Inventory 2011 (NEI11) and scaled to 88.9% to account for 2011–2013 $NO_x$ reductions (EPA, 2016); total emissions of NO for the domain are approximately $3.1 \times 10^6$ kg NO day$^{-1}$. The MEGAN model (Guenther et al., 2006) is used to determine biogenic emissions. Chemical initial and boundary conditions for the domain are obtained from the MOZART chemical model (Emmons et al., 2010). The RACM2 (Goliff et al., 2013) and MADE-SORGAM schemes are used to simulate gas-phase and aerosol chemistry respectively; the RACM2 scheme is customized to reflect recent advancements in understanding of alkyl nitrate chemistry using Browne et al. (2014) and Schwantes et al. (2015) as a basis. Lightning $NO_x$ emissions were inactive."

P7, L19: please clarify what the vertical range is of the 'first five layers of WRF-Chem.
Five model layers is approximately 500 m high, which is the same height used in Lu et al. (2015). The sentence now reads:

"The surface wind direction and speed are calculated as the average of the first five layers ($\sim 500$ m) of the 9 WRF 12 km grid cells closest to Atlanta at 1400 local standard time for each day."

P8, L6: is it really necessary or relevant to refer to a Matlab function, when explaining how you do the fit? If it is, please provide some more information on why you chose this particular fit approach over other alternatives. As a non-Matlab user, the sentence does not mean much to me. Later on the same for fmincon.
Other reviewers had similar comments; in the main paper, we've replaced references to Matlab functions with the mathematical algorithm behind it, which is an interior-point algorithm.

P8, 12-17: this part is very technical and should be moved to an appendix or supplement.
Again, other reviewers had similar comments. We have moved Table 1 and P8, L12–29 to the supplement. Section 2.5 now ends with "Technical details of the EMG fitting and uncertainty calculation are given in the supplement."

P10, L13: that lightning is not included in this WRF-Chem set-up should not be mentioned only here, but already in the model description section 2.3.
We have added the following sentence on P. 6, L. 4: "Lightning $NO_x$ emissions were inactive," (see end of change for previous comment on WRF-Chem details).

P12, L18: sigma_x represents the width of the Gaussian plume, but also the spatial smearing of the signal caused by the satellite pixel extent, and the fact that cities are covered by different satellite footprints from day to day.

Originally we had focused on its representation of the plume width because that is the most physically intuitive quantity, but we have extended the first sentence of P12, L18 to read:

> "$\sigma_x$ is the Gaussian smoothing length scale, representing both the width of the upwind Gaussian plume and smoothing of the $NO_2$ signal due to the physical extent of the source, the averaging of $NO_2$ within one OMI pixel, and daily variability in the overpass track (Beirle et al., 2011)."

P13, L14-19: this whole section presumably discusses Table 4, but that is not obvious from the text. I'm confused by the statement that the choice of a priori leads to statistically different emissions for all five cases, whereas Table 4 shows emission values that all overlap within the quoted uncertainty estimates.

When comparing two measured values, $t$-tests designed for either comparing replicate measurements or paired tests must be used (Harris, 2010). This paragraph is discussing results from using $t$-tests for comparing replicate measurements, i.e. two sample $t$-tests (Harris 2010, sect. 4-3, case 2). We chose this method because the emissions and lifetime estimates are the result of averaging VCDs over 3 months, so the fitting parameters are effectively means. Additionally, unlike the paired $t$-tests, this method takes into account the uncertainty in each value. Using these tests, any pair of emissions derived using different a priori for the same city and wind division are statistically different, even though the confidence intervals overlap.

To clarify which $t$-tests were used, we have modified P.13 L.14 to read ("2-sample" added):

> "We also use 2-sample $t$-tests at the 95% confidence level (Harris, 2010) to determine if differences in emissions and lifetimes given in Table 4 are significantly different..."

**References**

Acarreta, J. R., De Haan, J. F., and Stammes, P.: Cloud pressure retrieval using the O2-O2 absorption band at 477 nm, J. Geophys. Res. Atmos., 109, doi:10.1029/2003JD003915, URL http://dx.doi.org/10.1029/2003JD003915, d05204, 2004.

Beirle, S., Boersma, K., Platt, U., Lawrence, M., and Wagner, T.: "Megacity Emissions and Lifetimes of Nitrogen Oxides Probed from Space", Science, 333, 1737–1739, 2011.

Belmonte Rivas, M., Veefkind, P., Eskes, H., and Levelt, P.: OMI tropospheric $NO_2$ profiles from cloud slicing: constraints on surface emissions, convective transport and lightning $NO_x$, Atmos. Chem. Phys., 15, 13 519–13 553, doi:10.5194/acp-15-13519-2015, URL http://www.atmos-chem-phys.net/15/13519/2015/, 2015.

Boersma, K., Bucsela, E., Brinksma, E., and Gleason, J.: NO$_2$, in: OMI Algorithm Theoretical Basis Document, vol 4, OMI Trace Gas Algorithms, ATB-OMI-04, version 2.0, pp. 13–36, URL `http://eospso.nasa.gov/sites/default/files/atbd/ATBD-OMI-04.pdf`, 2002.

Boersma, K., Eskes, H., and Brinksma, E.: "Error analysis for tropospheric NO$_2$ retrieval from space, J. Geophys. Res. Atmos., 106, D04 311, doi:10.1029/2003JD003962, 2004.

Boersma, K., Eskes, H., Dirksen, R., van der A, R., Veefkind, J., Stammes, P., Huijnen, V., Kleipool, Q., Sneep, M., Claas, J., Leitão, J., Richter, A., Zhou, Y., and Brunner, D.: "An improved tropospheric NO$_2$ column retrieval algorithm for the Ozone Monitoring Instrument, Atmos. Meas. Tech., 4, 1905–1928, doi:10.5194/amt-4-1905-2011, 2011.

Boersma, K. F., Eskes, H. J., Veefkind, J. P., Brinksma, E. J., van der A, R. J., Sneep, M., van den Oord, G. H. J., Levelt, P. F., Stammes, P., Gleason, J. F., and Bucsela, E. J.: Near-real time retrieval of tropospheric NO¡sub¿2¡/sub¿ from OMI, Atmos. Chem. Phys., 7, 2103–2118, doi:10.5194/acp-7-2103-2007, 2007.

Browne, E. C., Wooldridge, P. J., Min, K.-E., and Cohen, R. C.: On the role of monoterpene chemistry in the remote continental boundary layer, Atmos. Chem. Phys., 14, 1225–1238, doi:10.5194/acp-14-1225-2014, 2014.

Bucsela, E., Krotkov, N., Celarier, E., Lamsal, L., Swartz, W., Bhartia, P., Boersma, K., Veefkind, J., Gleason, J., and Pickering, K.: "A new tropospheric and stratospheric NO$_2$ retrieval algorithm for nadir-viewing satellite instruments: applications to OMI, Atmos. Meas. Tech., 6, 2607–2626, doi:10.5194/amt-6-2607-2013, 2013.

Bucsela, E. J., Celarier, E. A., Wenig, M. O., Gleason, J. F., Veefkind, J. P., Boersma, K. F., and Brinksma, E. J.: Algorithm for NO$_2$ vertical column retrieval from the ozone monitoring instrument, IEEE T. Geosci. Remote, 44, 1245–1258, doi:10.1109/TGRS.2005.863715, 2006.

Burrows, J., Weber, M., Buchwitz, M., Rozanov, V., Ladstätter-Weißenmayer, A., Richter, A., DeBeek, R., Hoogan, R., Bramstedt, K., Eichmann, K.-U., Eisinger, M., and Perner, D.: The Global Ozone Monitoring Experiment (GOME): Mission Concept and First Scientific Results, 56, 151–175, doi:10.1175/1520-0469(1999)056⟨0151:TGOMEG⟩2.0.CO;2, 1999.

Choi, S., Joiner, J., Choi, Y., Duncan, B. N., Vasilkov, A., Krotkov, N., and Bucsela, E.: First estimates of global free-tropospheric NO$_2$ abundances derived using a cloud-slicing technique applied to satellite observations from the Aura Ozone Monitoring Instrument (OMI), Atmos. Chem. Phys., 14, 10 565–10 588, doi:10.5194/acp-14-10565-2014, URL `http://www.atmos-chem-phys.net/14/10565/2014/`, 2014.

Emmons, L. K., Walters, S., Hess, P. G., Lamarque, J.-F., Pfister, G. G., Fillmore, D., Granier, C., Guenther, A., Kinnison, D., Laepple, T., Orlando, J., Tie, X., Tyndall, G., Wiedinmyer, C., Baughcum, S. L., and Kloster, S.: Description and evaluation of the Model for Ozone and Related chemical Tracers, version 4 (MOZART-4), Geosci. Model

Dev., 3, 43–67, doi:10.5194/gmd-3-43-2010, URL `http://www.geosci-model-dev.net/3/43/2010/`, 2010.

EPA: Air Pollutant Emissions Trends Data, URL `https://www.epa.gov/air-emissions-inventories/air-pollutant-emissions-trends-data`, 2016.

Goliff, W. S., Stockwell, W. R., and Lawson, C. V.: The regional atmospheric chemistry mechanism, version 2, Atmos. Environ., 68, 174 – 185, doi:10.1016/j.atmosenv.2012.11.038, 2013.

Grell, G. A., Peckham, S. E., Schmitz, R., McKeen, S. A., Frost, G., Skamarock, W. C., and Eder, B.: Fully coupled online chemistry within the {WRF} model, Atmos. Environ., 39, 6957 – 6975, doi:10.1016/j.atmosenv.2005.04.027, 2005.

Guenther, A., Karl, T., Harley, P., Wiedinmyer, C., Palmer, P. I., and Geron, C.: Estimates of global terrestrial isoprene emissions using MEGAN (Model of Emissions of Gases and Aerosols from Nature), Atmos. Chem. Phys., 6, 3181–3210, doi:10.5194/acp-6-3181-2006, URL `http://www.atmos-chem-phys.net/6/3181/2006/`, 2006.

Harris, D.: Comparison of Means with Student's $t$, chap. 4-3, pp. 76–78, W.H. Freeman, 8th edn., 2010.

Koelemeijer, R. B. A. and Stammes, P.: Effects of clouds on ozone column retrieval from GOME UV measurements, J. Geophys. Res. Atmos., 104, 8281–8294, doi:10.1029/1999JD900012, 1999.

Lin, J.-T., Liu, M.-Y., Xin, J.-Y., Boersma, K. F., Spurr, R., Martin, R., and Zhang, Q.: Influence of aerosols and surface reflectance on satellite $NO_2$ retrieval: seasonal and spatial characteristics and implications for $NO_x$ emission constraints, Atmos. Chem. Phys., 15, 11 217–11 241, doi:10.5194/acp-15-11217-2015, 2015.

Lu, Z., Streets, D., de Foy, B., Lamsal, L., Duncan, B., and Xing, J.: "Emissions of nitrogen oxides from US urban areas: estimation from Ozone Monitoring Instrument retrievals for 2005–2014", Atmos. Chem. Phys., 15, 10 367–10 383, doi:10.5194/acp-15-10367-2015, 2015.

Marchenko, S., Krotkov, N. A., Lamsal, L. N., Celarier, E. A., Swartz, W. H., and Bucsela, E. J.: Revising the slant column density retrieval of nitrogen dioxide observed by the Ozone Monitoring Instrument, J. Geophys. Res. Atmos., 120, 5670–5692, doi:10.1002/2014JD022913, 2014JD022913, 2015.

Martin, R. V., Chance, K., Jacob, D. J., Kurosu, T. P., Spurr, R. J. D., Bucsela, E., Gleason, J. F., Palmer, P. I., Bey, I., Fiore, A. M., Li, Q., Yantosca, R. M., and Koelemeijer, R. B. A.: An improved retrieval of tropospheric nitrogen dioxide from GOME, J. Geophys. Res. Atmos., 107, doi:10.1029/2001JD001027, 4437, 2002.

McLinden, C. A., Fioletov, V., Boersma, K. F., Kharol, S. K., Krotkov, N., Lamsal, L., Makar, P. A., Martin, R. V., Veefkind, J. P., and Yang, K.: Improved satellite retrievals

of NO$_2$ and SO$_2$ over the Canadian oil sands and comparisons with surface measurements, Atmos. Chem. Phys., 14, 3637–3656, doi:10.5194/acp-14-3637-2014, 2014.

Schwantes, R. H., Teng, A. P., Nguyen, T. B., Coggon, M. M., Crounse, J. D., St. Clair, J. M., Zhang, X., Schilling, K. A., Seinfeld, J. H., and Wennberg, P. O.: Isoprene NO$_3$ Oxidation Products from the RO$_2$ + HO$_2$ Pathway, J. Phys. Chem. A, 119, 10 158–10 171, doi:10.1021/acs.jpca.5b06355, 2015.

Sneep, M., de Haan, J. F., Stammes, P., Wang, P., Vanbauce, C., Joiner, J., Vasilkov, A. P., and Levelt, P. F.: Three-way comparison between OMI and PARASOL cloud pressure products, J. Geophys. Res. Atmos., 113, doi:10.1029/2007JD008694, d15S23, 2008.

van Geffen, J. H. G. M., Boersma, K. F., Van Roozendael, M., Hendrick, F., Mahieu, E., De Smedt, I., Sneep, M., and Veefkind, J. P.: Improved spectral fitting of nitrogen dioxide from OMI in the 405465 nm window, Atmos. Meas. Tech., 8, 1685–1699, doi: 10.5194/amt-8-1685-2015, 2015.

Vinken, G. C. M., Boersma, K. F., van Donkelaar, A., and Zhang, L.: Constraints on ship NO$_x$ emissions in Europe using GEOS-Chem and OMI satellite NO$_2$ observations, Atmos. Chem. Phys., 14, 1353–1369, doi:10.5194/acp-14-1353-2014, URL `http://www.atmos-chem-phys.net/14/1353/2014/`, 2014.

Ziemke, J., Chandra, S., and Bhartia, P.: "Cloud slicing: A new technique to derive upper tropospheric ozone from satellite measurements", J. Geophys. Res. Atmos., 106, 9853–9867, 2001.

Ziemke, J. R., Joiner, J., Chandra, S., Bhartia, P. K., Vasilkov, A., Haffner, D. P., Yang, K., Schoeberl, M. R., Froidevaux, L., and Levelt, P. F.: Ozone mixing ratios inside tropical deep convective clouds from OMI satellite measurements, Atmos. Chem. Phys., 9, 573–583, doi:10.5194/acp-9-573-2009, 2009.